# Improving group robustness under noisy labels using predictive uncertainty

## Abstract

The standard empirical risk minimization (ERM) can underperform on certain minority groups (i.e., waterbirds in lands or landbirds in water) due to the spurious correlation between the input and its label. Several studies have improved the worst-group accuracy by focusing on the high-loss samples. The hypothesis behind this is that such high-loss samples are *spurious-cue-free* (SCF) samples. However, these approaches can be problematic since the high-loss samples may also be samples with noisy labels in the real-world scenarios. To resolve this issue, we utilize the predictive uncertainty of a model to improve the worst-group accuracy under noisy labels. To motivate this, we theoretically show that the high-uncertainty samples are the SCF samples in the binary classification problem. This theoretical result implies that the predictive uncertainty is an adequate indicator to identify SCF samples in a noisy label setting. Motivated from this, we propose a novel ENtropy based Debiasing (END) framework that prevents models from learning the spurious cues while being robust to the noisy labels. In the END framework, we first train the *identification model* to obtain the SCF samples from a training set using its predictive uncertainty. Then, another model is trained on the dataset augmented with an oversampled SCF set. The experimental results show that our END framework outperforms other strong baselines on several real-world benchmarks that consider both the noisy labels and the spurious-cues.

## 1 Introduction

The standard Empirical Risk Minimization (ERM) has shown a high error on specific groups of data although it achieves the low test error on the in-distribution datasets. One of the reasons accounting for such degradation is the presence of *spurious-cues*. The spurious cue refers to the feature which is highly correlated with labels on certain training groups—thus, easy to learn—but not correlated with other groups in the test set (Nagarajan et al., 2020; Wiles et al., 2022). This spurious-cue is problematic especially occurs when the model cannot classify the minority samples although the model can correctly classify the majority of the training samples using the spurious cue. In practice, deep neural networks tend to fit easy-to-learn simple statistical correlations like the spurious-cues (Geirhos et al., 2020). This problem arises in the real-world scenarios due to various factors such as an observation bias and environmental factors (Beery et al., 2018; Wiles et al., 2022). For instance, an object detection model can predict an identical object differently simply because of the differences in the background(Ribeiro et al., 2016; Dixon et al., 2018; Xiao et al., 2020).

In nutshell, there is a low accuracy problem caused by the spurious-cues being present in a certain group of data. In that sense, importance weighting (IW) is one of the classical techniques to resolve this problem. Recently, several deep learning methods related to IW (Sagawa et al., 2019; 2020; Liu et al., 2021; Nam et al., 2020) have shown a remarkable empirical success. The main idea of those IW-related methods is to train a model with using data oversampled with hard (high-loss) samples. The assumption behind such approaches is that the high-loss samples are free from the spurious cues because these shortcut features generally reside mostly in the low-loss samples Geirhos et al. (2020). For instance, Just-Train-Twice (JTT) trains a model using an oversampled training set containing the *error set* generated by the *identification model*.

On the other hand, noisy labels are another factor of performance degradation in the real-world scenario. Noisy labels commonly occur in massive-scale human annotation data, biology and chem-

istry data with inevitable observation noise (Lloyd et al., 2004; Ladbury & Arold, 2012; Zhang et al., 2016). In practice, the proportions of incorrectly labeled samples in the real-world human-annotated image datasets can be up to 40% (Wei et al., 2021). Moreover, the presence of noisy labels can lead to the failure of the high-loss-based IW approaches, since a large value of the loss indicates not only that the sample may belong to a minority group but also that the label may be noisy (Ghosh et al., 2017). In practice, we observed that even a relatively small noise ratio (10%) can impair the high-loss-based methods on the benchmarks with spurious-cues, such as Waterbirds and CelebA. This is because the high loss-based approaches tend to focus on the noisy samples without focusing on the minority group with spurious cues.

Our observation motivates the principal goal of this paper: how can we better select only spurious-cue-free (SCF) samples while excluding the noisy samples? As an answer to this question, we propose the **predictive uncertainty-based sampling** as an oversampling criterion, which outperforms the error-set-based sampling. The predictive uncertainty has been used to discover the minority or unseen samples (Liang et al., 2017; Van Amersfoort et al., 2020). We utilize such uncertainty to detect the SCF samples. In practice, we train the *identification model* via the noise-robust loss and the Bayesian neural network framework to obtain reliable uncertainty for the minority group samples. By doing so, the proposed *identification model* is capable of properly identifying the SCF sample while preventing the noisy labels from being focused on. After training the identification model, similar to JTT, the *debiased model* is trained with the SCF set oversampled dataset. Our novel framework, ENtropy-based Debiasing (END), shows an impressive worst-group accuracy on several benchmarks with various degrees of symmetric label noise. Furthermore, as a theoretical motivation, we demonstrate that the predictive uncertainty (entropy) is a proper indicator for identifying the SCF set regardless of the existence of the noisy labels in the simple binary classification problem setting.

To summarize, our key contributions are three folds:

1. We propose a novel predictive uncertainty-based oversampling method that effectively selects the SCF samples while minimizing the selection of noisy samples.
2. We rigorously prove that the predictive uncertainty is an appropriate indicator for identifying a SCF set in the presence of the noisy labels, which well supports the proposed method.
3. We propose additional model considerations for real-world applications in both classification and regression tasks. The overall framework shows superior worst-group accuracy compared to recent strong baselines in various benchmarks.

## 2 RELATED WORKS

**Noisy label robustness: small loss samples**     In this paper, we focus on two types of the noisy label robustness studies: **(1)** a sample re-weighting based approach and **(2)** a robust loss functions based approach. First, the sample re-weighting methods assign sample weights during model training to achieve the robustness against the noisy label (Han et al., 2018; Ren et al., 2018; Wei et al., 2020; Yao et al., 2021). Alternatively, the robust loss function based approaches design the loss function which implicitly focuses on the clean label (Reed et al., 2015; Zhang & Sabuncu, 2018; Thulasidasan et al., 2019; Ma et al., 2020). The common premise of the *sample re-weighting* and *robust loss function* methods are that the low-loss samples are likely to be the clean samples. For instance, Co-teaching uses two models which select the clean sample for each model by choosing samples of small losses (Han et al., 2018). Similarly, (Zhang & Sabuncu, 2018) design the generalized cross entropy loss function to have less emphasis on the samples of large loss than the vanilla cross entropy.

**Group robustness: large loss samples**     The model with the *group robustness* should yield a low test error regardless of the group specific information of samples (i.e., groups by background images). This group robustness can be improved if the model does not focus on the spurious cues (i.e., the background). The common assumption of prior works on the group robustness is that the large loss samples are spurious-cue-free. Sagawa et al. (2019); Zhang et al. (2020) propose the Distributionally Robust Optimization (DRO) methods which directly minimize the worst-group loss via group information of the training datasets given a priori. On the other hand, the group information-free approaches (Namkoong & Duchi (2017); Arjovsky et al. (2019); Oren et al. (2019)) have been proposed due to the non-negligible cost of group information. These approaches aim at achieving the

group robustness by training the model via an implicit worst-case loss (e.g., maximum loss around the current loss). Nam et al. (2020) trains the *debiased* model by focusing on the samples with a large-loss with respect to the *biased* model. Notably, Just-Train-Twice (JTT) by Liu et al. (2021) is simple but effective framework. In JTT, an *identification model* is trained to build an *error set*, which consists of the wrongly predicted samples by the identification model. Then, the *final model* is trained after adding the oversampled error set into the original training dataset.

## 3 PRELIMINARY: WORST GROUP PERFORMANCE AND SPURIOUS CUE

We consider a supervised learning task with inputs $\mathbf{x}^{(i)} \in \mathcal{X}$, corrupted labels $y^{(i)} \in \mathcal{Y}$, and true labels $z^{(i)} \in \mathcal{Y}$. We let a latent attribute of corresponding $\mathbf{x}^{(i)}$ as $a^{(i)} \in \mathcal{A}$ (e.g., different backgrounds in Figure 1). We assume each triplet $(\mathbf{x}^{(i)}, y^{(i)}, z^{(i)})$ belongs to a corresponding group, $g^{(i)} \in \mathcal{G}$ (e.g., group by a background attribute and its true class $(z^{(i)}, a^{(i)})$ in Figure 1). We also denote a training dataset as $D = \{(\hat{\mathbf{x}}^{(i)}, \hat{y}^{(i)})\}_{i=1}^{N}$ where each pair $(\hat{\mathbf{x}}^{(i)}, \hat{y}^{(i)})$ is sampled from a data distribution $D^*$ on $\mathcal{X} \times \mathcal{Y}$. Importantly, an attribute $a^{(i)}$ can be spuriously correlated with a label on a certain dataset $D$. For instance, in Figure 1, the label (cow/camel) can be highly correlated with the background feature (green pasture/desert), implying a false causal relationship. In this case, the background feature is a *spurious cue*.

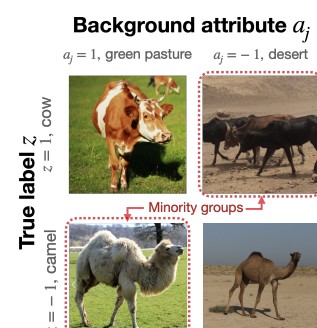

**Background attribute** $a_j$

$a_j = 1$, green pasture       $a_j = -1$, desert

True label $z$     $z = 1$, cow     $z = -1$, camel

Minority groups

Figure 1: The example case of the minority groups and its attribute on the cow-camel classification problem.

Ideally, we aim to minimize the worst-group risk of a model $f_\theta$ : $\mathcal{X} \to \mathbb{R}^c$ ($c$ is the number of class), parameterized by $\theta$, with an unknown data distribution $D^*$ and true labels $z$ as follows:

$$\theta^* = \arg\min_\theta [\max_{g \in \mathcal{G}} R_{D_g^*}(\theta)], \qquad R_{D_g^*}(\theta) = \mathbb{E}_{(\mathbf{x}, z) \sim D_g^*}[L(f_\theta(\mathbf{x}), z)] \qquad (1)$$

where $D_g^*$ is the data distribution for the group $g$ and $L : \mathbb{R}^c \times \mathcal{Y} \to \mathbb{R}$ is a loss function. To achieve the above goal of improving the worst group accuracy, **the prediction of the model $f_\theta$ should not depend on the spurious cues.** We instantiate this with the cow or camel classification dataset (Beery et al., 2018) which includes the background features as the spurious cue. When this relationship is abused, a model can easily classify the majority groups while failing to classify the minority groups such as a cow in the desert. Thus, the model has a poor accuracy on the minority group, leading to a high worst-group error.

To improve the worst-group accuracy, it is ideal to directly optimize the model via Eq 1. However, in the real-world, we assume that only the training samples $D$ are given, with **no information about groups $g$, attributes $a$, and true labels $z$ during training.** Therefore, ERM with the corrupted label is the alternative solution for the parameter $\theta$:

$$\theta_D^* = \arg\min_\theta \hat{R}_D(\theta), \qquad \hat{R}_D(\theta) = \frac{1}{N} \sum_{i=1}^{N} L(f_\theta(\hat{\mathbf{x}}^{(i)}), \hat{y}^{(i)}) \qquad (2)$$

Our goal in this paper is to resolve the problem caused by the unavoidable alternative, ERM. The problem lies in the *poor accuracy on a specific group* due to the model's reliance on the spurious cues (Sagawa et al., 2019). To address the problem of the poor worst-group accuracy, the common assumption in the literature has been that the model training should focus more on the particular samples that are not classifiable using the spurious cues (e.g., oversampling the cow on the desert samples) (Sagawa et al., 2019; Xu et al., 2020; Liu et al., 2021). In this paper, we call such samples the *Spurious-Cue-Free* (SCF) samples.

Given that we lack information to determine which samples are SCF, the remaining question is "*how can we identify which samples belong to a SCF set?*". The previous studies (Liu et al., 2021) hypothesize that the samples with large loss values correspond to the SCF set (Sec 2). These approaches, however, have limitations because large loss values can be attributed to both the SCF and the noisy samples. Going a step further than other approaches, **our primary strategy for identifying a SCF set is obtaining samples with a high predictive uncertainty.** By doing so, our approach allows a more careful selection of SCF samples while excluding noisy ones as much as possible. As theoretical support, we rigorously show in the following section that the predictive uncertainty is a proper indicator to identify a SCF set in the presence of the noisy labels.

## 4 GROUP ROBUSTNESS UNDER NOISY LABELS VIA UNCERTAINTY

In this section, we primarily show that **the predictive uncertainty is a proper indicator to identify the SCF samples under the noisy label environment.** Not only that, we can verify that utilizing the loss values as an oversampling metric can fail to distinguish SCF samples from the noisy samples. To rigorously prove this, we theoretically analyze the binary classification problem including both the spurious cues and the noise.

**Problem setup: data distribution and model hypothesis**   Consider a $d$-dimensional input $\mathbf{x} = (x_1, \ldots, x_d)$ and its features and labels are binary: $x_i, y, z \in \{-1, 1\}$. A data generation procedure for triplets $(\mathbf{x}, y, z)$ is defined as following: first, $z$ is uniformly sampled over $\{-1, 1\}$. Then, if the true label is positive ($z = 1$), $\mathbf{x} \sim \mathcal{B}_{\mathbf{p}}$ where $\mathcal{B}_{\mathbf{p}}$ is a distribution of independent Bernoulli random samples and $\mathbf{p} = (p_1, \ldots, p_d) \in [0, 1]^d$. Here, $p_i$ represents the probability that the $i$-th feature has a value of 1. On the contrary, if $z = -1$, $\mathbf{x}$ is sampled from a different distribution: $\mathbf{x} \sim \mathcal{B}_{\mathbf{p}'}$ where $\mathbf{p}' = (p'_1, \ldots, p'_d) \in [0, 1]^d$. Furthermore, with the probability $\eta$, there is a label noise: $y = -z$. Otherwise, $y = z$. Finally, we consider a linear model with parameters $\beta = (\beta_0, \ldots \beta_d)$, which are optimized via risk minimization over the joint distribution of the features and noisy labels $(\mathbf{x}, y)$. This problem setup is inspired by the problem definition of Nagarajan et al. (2020) and Sagawa et al. (2020), representing the spurious features.

Next, the concept of the spurious cue in the defined classification task is demonstrated using the cow and camel images shown in Figure 1. Let's assume that the $j$'th feature represents a background attribute ($x_j = a$), which is either a green pasture ($x_j = 1$) or a desert ($x_j = -1$). Suppose 98% of cows have a green pasture background (thus, $p_j = 0.98$) while only 5% of camels have the green pasture background ($p'_j = 0.05$). In this case, the majority of the data could likely be classified only using the $x_j$ feature (e.g., only utilizing $\beta_j$). However, abusing this spurious feature could hinder a model from accurately classifying the minority groups such as cows in a desert. Note that in practice, there can be many spurious features that correspond to a given latent attribute ($a$).

To quantify the spuriousness, we define the *Spurious-Cue Score* (SCS) function.

**Definition 1 (Spurious cue score function)**   *we define the spurious cue score function* $\Psi_{\mathbf{p},\mathbf{p}'}$ : $\mathbb{R}^d \to \mathbb{R}$ *with any function* $s(\cdot, \cdot)$ *which satisfies followings:*

$$\Psi_{\mathbf{p},\mathbf{p}'}(\mathbf{x}) = \sum_{i=1}^{d} s(p_i, p'_i)x_i, \quad \begin{cases} s(p_i, p'_i) > 0, & \text{If } p_i > p'_i \\ s(p_i, p'_i) \leq 0, & \text{If } p_i \leq p'_i \end{cases} \tag{3}$$

Intuitively, if the SCS function value is low, the model will struggle to correctly predict the label of a sample $\mathbf{x}$ by relying solely on features having a high correlation with the label. A simple example of $s(\cdot, \cdot)$ can be $s(p_j, p'_j) = p_j - p'_j$. In the cow/camel classification problem, the cow in the desert sample has the lower $\Psi_{\mathbf{p},\mathbf{p}'}$ value due to the term $s(p_j, p'_j)x_j = (0.98 - 0.05)(-1) = -0.93$. In contrast, the majority of cow has 0.93. For the negative class samples (camel), $\Psi_{\mathbf{p}',\mathbf{p}}$ can be used instead. Importantly, the SCS function is only determined by the true labels ($z$) and the input features ($\mathbf{x}$) but *not the noisy labels* ($y$).

With Definition 1, the following result formalizes our goal: to find the SCF samples (having a low SCS function values) by obtaining samples with the high predictive uncertainty.

**Theorem 1**   *Given any* $0 < \epsilon < 1/2$, *for a sufficiently small* $\delta$, *a sufficiently large* $d$ *and any* $\mathbf{p}, \mathbf{p}' \in [\epsilon, 1 - \epsilon]^d$ *with* $|p_i - p'_i| \geq \epsilon$ $(1 \leq i \leq d)$, *the following holds for any* $0 \leq \eta < 1/2$:

$$\mathbb{P}_{\mathbf{x},y,z}[\ R(z, \mathbf{x}) \leq \epsilon \mid F(H_{\beta^*}(\mathbf{x})) \geq 1 - \delta \ ] \geq 1 - \epsilon,$$
$$R(\mathbf{x}, z) = \mathbf{1}_{z=1}F_{\mathbf{x}\sim\mathcal{B}_{\mathbf{p}}}(\Psi_{\mathbf{p},\mathbf{p}'}(\mathbf{x})) + \mathbf{1}_{z=-1}F_{\mathbf{x}\sim\mathcal{B}_{\mathbf{p}'}}(\Psi_{\mathbf{p}',\mathbf{p}}(\mathbf{x})) \tag{4}$$

*where* $\beta^*$ *is the risk minimization solution of the linear regression on the distribution on* $(\mathbf{x}, y)$ *and* $H_{\beta^*}$ *is the predictive entropy of the model with its parameter* $\beta^*$. *F*, $F_{\mathbf{x}\sim\mathcal{B}_{\mathbf{p}}}$ *and* $F_{\mathbf{x}\sim\mathcal{B}_{\mathbf{p}'}}$ *are the cumulative distribution functions with respect to the data distribution,* $\mathcal{B}_{\mathbf{p}}$ *and* $\mathcal{B}_{\mathbf{p}'}$ *respectively.*

Theorem 1 states that the highly uncertain samples ($F(H_{\beta^*}(\mathbf{x})) \geq 1 - \delta$) are more likely to have a low SCS function values among samples belonging to the same class ($R(\mathbf{x}, z) \leq \epsilon$). Thus, this

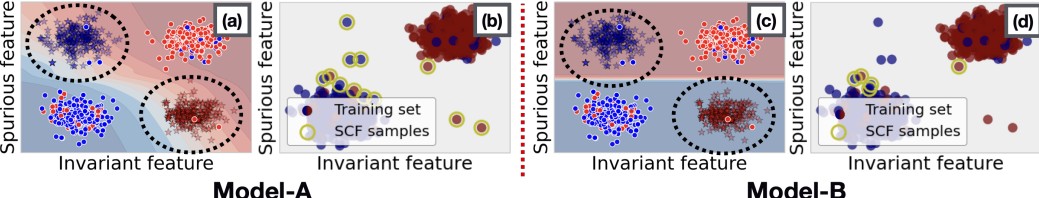

Figure 2: 2-D classification results of *identification models* on synthetic data. The blue/red represent classes. The deeper the background, the more confident the prediction. The dots and translucent stars represent training and test data, respectively. Here, *true* SCF samples are in dotted circles (cannot be classified via spurious feature). Two classification results (**(a)** and **(b)**) shows that *Model-A* (proposed) well identifies the SCF set (large overlaps between yellow circles in **(b)** and true SCF samples) while *Model-B* (without regularization) fails due to its overconfident uncertainty for the *true* SCF samples.

statement implies that utilizing the uncertainty (predictive entropy) can be useful in identifying the SCF samples. Additionally, it can be observed that the selectivity of the predictive entropy ($H_{\beta*}$) is independent of the presence of the label noise (whether $yz = -1$). Particularly, it's worth noting that the probability of the samples with a high predictive uncertainty to have a label noise is not greater than $\eta$, the original label noise ratio. On the contrary, the probability of the label noise to be present in the samples with high loss exceeds $\eta$ (Theorem 2 in Appendix A.6). We empirically demonstrate that the proposed framework with the predictive uncertainty outperforms the existing loss value-based ones in the real-world benchmarks with the noisy labels (Sec 6). The formal form of this theorem and its proof are in Appendix A.

**Entropy based debiasing for neural networks trained via ERM**   We theoretically show that utilizing the predictive uncertainty can allow us to obtain the SCF samples under the ideal condition. However, there are two obstacles in applying this to the real-world scenario. Firstly, an overparameterized neural network trained via ERM (with a finite number of samples) can perfectly *memorize* the label noises (Zhang et al., 2017). This *memorization* could cause samples with the noisy labels to have a high entropy during training. For instance, as the model reduces the loss of the noisy samples once after it has fitted to the noise-free samples, the predictive entropy of those noisy samples increases (Xia et al., 2020). Secondly, since the neural network models generally tend to be overconfident (Guo et al., 2017; Hein et al., 2019), the proposed framework could potentially be unreliable in entropy-based acquisition of the SCF samples.

To summarize, utilizing the predictive uncertainty with the neural networks necessitates two requirements for a neural network: **(1: robustness to label noise)** the model should not memorize samples with noisy labels; **(2: reliable uncertainty)** the model should not be overconfident when identifying the SCF samples. As an illustrative example, Figure 2 presents how well two different models identify the SCF samples. *Model-A* (proposed) is successful because it satisfies these two requirements. In contrast, *model-B* fails due to the overconfident prediction (low predictive uncertainty) for the true SCF samples. In the following section, we describe the proposed framework along with the training process that is designed to satisfy these two requirements.

## 5   ENTROPY BASED DEBIASING

**Overview**   The proposed ENtropy-based Debiasing (END) framework focuses on training the samples with high predictive uncertainty. The END framework, in particular, is made up of two models with identical architectures. First, the *identification model* uses the predictive uncertainty to identify the SCF samples. Second, the *debiased model* is trained on a newly constructed training data set by oversampling the SCF set. As a result, END achieves group and noise label robustness. In addition, our framework can also be extended to regression problems, whereas other baselines cannot.

## 5.1 IDENTIFICATION MODEL

Robustness to the label noise and reliable uncertainty are two major requirements for using the predictive uncertainty with neural networks. To achieve these goals, we design the proposed identification model with the loss function robust to noisy labels and the overconfidence regularizer. In addition, we employ a Bayesian Neural Network (BNN) to obtain reliable uncertainty. Additionally, we explain how to modify the proposed identification model to fit a regression task.

**The noise-robust loss function and the overconfidence regularizer**   For the loss function, we use the mean absolute error (MAE) loss instead of the typical cross-entropy loss because the noise-label robustness of MAE has been well demonstrated in the literature (Ghosh et al., 2017; Zhang & Sabuncu, 2018). The MAE loss $L_{MAE}$ is defined as the following:

$$L_{MAE}(f_\theta(\hat{\mathbf{x}}), \hat{\mathbf{y}}) = \|\hat{\mathbf{y}} - \sigma(f_\theta(\hat{\mathbf{x}}))\|_1 = 2 - 2\sigma_{i*}(f_\theta(\hat{\mathbf{x}})) \tag{5}$$

where $i^*$ is the index at which $\hat{\mathbf{y}}_{i*} = 1$ in the one-hot encoded $\hat{\mathbf{y}}$ and $\sigma_i(\cdot)$ is the $i$-th value of the softmax function. Although a model with the MAE loss alone could generally be noise-robust, it may occasionally be overconfident in predicting the SCF samples, meaning that the model produces low uncertainties in its predictions of those samples. As a result, the framework decides to exclude them from the SCF set. This problem is visually demonstrated in *Model-B* of Figure 2, which is trained with the MAE alone. To resolve this, we employ the confidence regularization. Importantly, the role of this regularization is to prevent overconfident prediction for the SCF samples (Liang et al., 2018; Müller et al., 2019; Utama et al., 2020). Specifically, this confidence regularization (Pereyra et al., 2017) penalizes the predictive entropy. The regularizer $R_{ent}$ is defined as follows:

$$R_{ent}(f_\theta(\hat{\mathbf{x}})) = \sum_{i=1}^c \sigma_i(f_\theta(\hat{\mathbf{x}})) \log(\sigma_i(f_\theta(\hat{\mathbf{x}}))) \tag{6}$$

In practice, we empirically show that combining MAE with the confidence regularization yields a better set of SCF samples (*Model-A* in Figure 2), which enhances the worst-group accuracy (Sec 6.1 and 6.2). In addition, the ablation study (Appendix C.1) shows that the contribution of their combination is significant on the classification benchmarks.

**Bayesian neural network**   We chose BNN as a network architecture to ensure that the identification model's uncertainty is reliable. This identification model is trained using the widely used stochastic gradient Markov-chain Monte Carlo sampling algorithm, Stochastic Gradient Langevin Dynamics (SGLD) (Welling & Teh, 2011). The SGLD updates the parameter $\theta_t$ with the batch $\{\hat{\mathbf{x}}^{(i)}, \hat{y}^{(i)}\}_{i=1}^n$ at step $t$ via the following equation:

$$\theta_{t+1} \leftarrow \theta_t - \left[ -\frac{\epsilon_t}{2}(\nabla_\theta \log p(\theta_t) - \frac{N}{n}\sum_{i=1}^n \nabla_\theta \log p(\hat{y}^{(i)}|\theta_t, \hat{\mathbf{x}}^{(i)})) + \rho_t \right] \tag{7}$$

where $\epsilon_t$ is the step size and $\rho_t \sim \mathcal{N}(0, \epsilon_t)$. The negative log-likelihood term $(-\log p(\hat{y}^{(i)}|\theta_t, \hat{\mathbf{x}}^{(i)}))$ can be interpreted as a loss function. The prior term $(\log p(\theta_t))$ is equivalent to the L2 regularization if we use the Gaussian prior over the $\theta$. During the parameter updates, we obtain the snapshots of the parameters at every $K$ steps. The final predictive value of the identification model is the empirical mean of the predictions: $\tilde{f}(\mathbf{x}) = \frac{1}{M}\sum_{j=1}^M f_{\theta^{(j)}}(\mathbf{x})$ where $M$ is the number of parameter snapshots, $\theta^{(j)}$ is the parameters at the $j$'th snapshot.

**Extension to regression**   Another benefit of the END framework is that it can be easily extended to fit a regression task with minor modifications. Since we utilize the BNN, the entropy can be calculated over a gaussian distribution with a predictive mean and a variance of SGLD weight samples $\mathcal{N}(\tilde{f}(\mathbf{x}), \text{Var}_j[f_{\theta^{(j)}}(\mathbf{x})])$ (Kendall & Gal, 2017). Instead of the classification-MAE loss (Eq 5), the regression version of the identification model uses the common regression-MAE loss $(L(\mathbf{x}, y) = |f(\mathbf{x}) - y|)$ similar to the classification task. Another change in the regression version is that the confidence regularization is no longer used because it is not defined in the regression task. The regression version of END also improves worst-group performance in the regression task as shown in Sec 6.3.

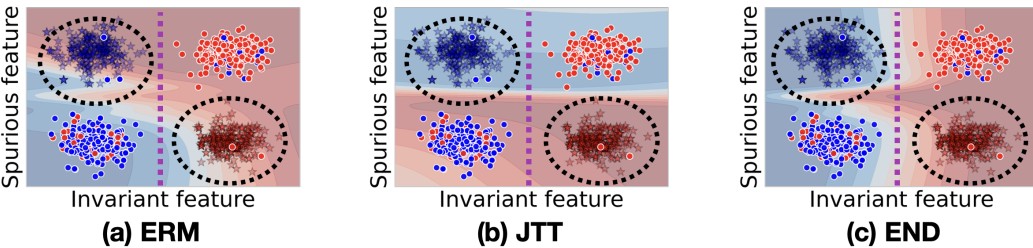

Figure 3: 2-D classification results on the synthetic data. The blue/red represents the classes. The dots and translucent stars represents training and test data respectively. The minority (SCF) groups are in the dotted circles. The ideal decision boundary is the vertical purple dotted lines. The decision boundary of END is the closest to the ideal boundary.

## 5.2 DEBIASED MODEL

Once the identification model is trained, we build a new training dataset by oversampling the SCF samples to train the debiased model. First, we assume that the predictive entropy of the identification model follows the Gamma distribution because it is a common assumption that the variance (uncertainty) of the Gaussian distribution follows the Gamma distribution (Bernardo & Smith, 2009). Then, we obtain the SCF samples based on the p-value cut-off in a fitted Gamma distribution. Formally, the SCF set ($D_{SCF}^k$) obtained by the identification model is as follows:

$$D_{SCF}^k = \underbrace{\hat{D}_\tau \cup \cdots \cup \hat{D}_\tau}_{k \text{ times}}, \quad \hat{D}_\tau = \{(\hat{\mathbf{x}}^{(i)}, \hat{y}^{(i)}) | \Phi(H(\sigma(\tilde{f}(\hat{\mathbf{x}}^{(i)})); \alpha^*, \beta^*) > 1 - \tau\} \quad (8)$$

where $H(\cdot)$ is entropy; $\Phi$ is the CDF of the Gamma distribution; $\tau$ is the p-value threshold[1]; $k$ is the hyperparameter to represent the degree of oversampling. The parameters of the gamma distribution, $\alpha^*$ and $\beta^*$, are fitted via the moment estimation method (Hansen, 1982).

Finally, the *debiased model* is trained via ERM with the new dataset $\hat{D} \cup D_{SCF}^k$ after acquisition of the SCF set via eq 8. Note that training the debiased model follows a conventional ERM procedure: it does not use the confidence regularization or the MAE loss. The final prediction of the END framework is the predictive value of the trained debiased model.

## 6 EXPERIMENTS

### 6.1 SYNTHETIC DATASET EXPERIMENT

We begin with the 2D-classification synthetic dataset experiment to qualitatively substantiate the group and noise label robustness of the END framework. The dataset has two characteristics: it has (**1**) the spurious features and (**2**) the noisy labels. Initially, we describe the two features of the dataset. One feature is the spurious feature which is easy to learn but exploiting this feature cannot classify the test set (Figure 3). The other one is the invariant feature. This feature is hard to learn because we manually scale down this feature value. Only a few training samples' labels solely rely on this feature. The ideal model can correctly classify both the train and the test sets only when using the invariant features. Therefore, the ideal decision boundary will be the vertical line in Figure 3. Secondly, we assign random labels to the training samples with the probability of 20% to evaluate the model's robustness to the noisy labels. The details of the dataset and the neural network model are in Appendix B.

The experimental results show that END correctly classifies both the majority and the minority groups (Figure 3(c)). We posit that the outperformance is due to the well-identified SCF sample (Model-A in Figure 2). On the other hand, ERM is insufficient to classify the minority group (samples in the dotted circles), although it perfectly fits the majority group samples(Figure 3(a)). This poor performance of ERM is consistent with the empirical studies on the real-world datasets like Waterbirds and CelebA (Liu et al., 2021). Notably, JTT has learned a wrong decision boundary in favor of the minority group while completely overlooking the majority group. This is because JTT focuses too much on the noisy labels.

---

[1]Note the similarity between the definition of $\hat{D}_\tau$ and the condition $F(H_{\beta^*}(\mathbf{x})) \geq 1 - \delta$ in Theorem 1.

Table 1: Average accuracy (ACC) and worst-group accuracy (WG Acc) evaluated on the Waterbirds and CelebA dataset, with varying noise levels. END consistently outperforms other baselines, especially for the noisy datasets.

| Noise level | Clean | | 10% noise | | 20% noise | | 30% noise | |
|---|---|---|---|---|---|---|---|---|
| *Waterbirds* | WG Acc | Acc | WG Acc | Acc | WG Acc | Acc | WG Acc | Acc |
| ERM | 0.687 (0.01) | 0.969 (0.00) | 0.648 (0.03) | 0.945 (0.01) | 0.649 (0.05) | 0.913 (0.01) | 0.629 (0.06) | 0.893 (0.03) |
| ERM (GCE) | 0.674 (0.01) | 0.968 (0.00) | 0.665 (0.03) | 0.945 (0.00) | 0.651 (0.04) | 0.902 (0.00) | 0.660 (0.07) | 0.885 (0.03) |
| LfF | 0.710 (0.03) | 0.947 (0.02) | 0.710 (0.00) | 0.914 (0.03) | 0.726 (0.03) | 0.858 (0.04) | 0.660 (0.04) | 0.899 (0.02) |
| JTT | **0.846 (0.03)** | 0.865 (0.02) | 0.565 (0.08) | 0.670 (0.04) | 0.060 (0.03) | 0.135 (0.01) | 0.027 (0.01) | 0.107 (0.00) |
| END | 0.828 (0.01) | 0.934 (0.01) | **0.842 (0.01)** | 0.914 (0.01) | **0.832 (0.01)** | 0.887 (0.02) | **0.818 (0.01)** | 0.854 (0.01) |
| *CelebA* | WG Acc | Acc | WG Acc | Acc | WG Acc | Acc | WG Acc | Acc |
| ERM | 0.487 (0.03) | 0.952 (0.00) | 0.477 (0.01) | 0.927 (0.01) | 0.480 (0.02) | 0.891 (0.01) | 0.485 (0.01) | 0.858 (0.03) |
| ERM (GCE) | 0.502 (0.03) | 0.956 (0.00) | 0.524 (0.01) | 0.950 (0.00) | 0.522 (0.02) | 0.941 (0.00) | 0.526 (0.04) | 0.920 (0.01) |
| LfF | 0.788 (0.03) | 0.871 (0.04) | 0.080 (0.06) | 0.217 (0.01) | 0.027 (0.02) | 0.089 (0.02) | 0.052 (0.06) | 0.236 (0.25) |
| JTT | 0.822 (0.02) | 0.915 (0.01) | 0.748 (0.02) | 0.810 (0.01) | 0.245 (0.36) | 0.357 (0.29) | 0.151 (0.16) | 0.258 (0.12) |
| END | **0.826 (0.02)** | 0.889 (0.00) | **0.797 (0.01)** | 0.893 (0.01) | **0.811 (0.02)** | 0.901 (0.01) | **0.778 (0.03)** | 0.892 (0.01) |

## 6.2 GROUP ROBUSTNESS BENCHMARK DATASET

In this subsection, we evaluate the END framework on the two benchmark image datasets: the CelebA and Waterbirds, which have the spurious-cues (Wah et al., 2011; Liu et al., 2015). To evaluate **both the group** and the **noise label robustness**, we add simple symmetric label noises (uniformly flip the label) to the datasets, as shown in Table 1.

In this experiment, we use two kinds of baselines: the group-robust and the noise-robust baselines. The group-robust baselines include JTT (Liu et al., 2021), ERM (Zhang & Sabuncu, 2018), and Learning-from-Failure (LfF) (Nam et al., 2020). There is one noise-robust baseline, ERM with GCE (Generalized Cross Entropy) (Zhang & Sabuncu, 2018). We use the identical model architecture, ResNet50 He et al. (2016), for all baselines and END. For JTT, LfF, and ERM, we follow the identical experimental setup as Liu et al. (2021) presented. Details of the experimental setup are in Appendix B.

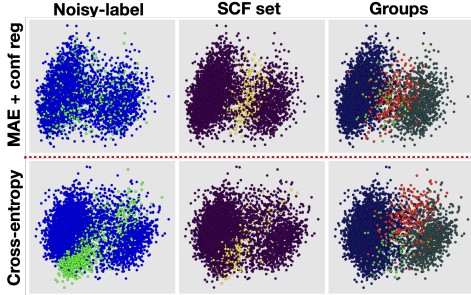

Figure 4: 2-D PCA projection of the latent feature of the identification model (first row) and its variant (w/o $L_{MAE}$ and $R_{ent}$) (second row). Each dot represents the training sample. (first column) The green dots are noisy labels, and the blue dots are clean labels; (second column) The yellow dots are the obtained SCF samples; (third column) Each color represents corresponding groups.

The results from Table 1 substantiate that, unlike the other baselines, END achieves **both** the **group** and the **noisy-label** robustness. The primary reason for this is that END employs the predictive entropy, which is unaffected by the label noise. Specifically, we observe that the worst-group accuracy of the END framework consistently outperforms the other models in the noisy cases (Table 1; $>= 10\%$ noise). Not only that, END also shows the competitive worst-group accuracy on the noise free case. On the other hand, as the noise rate increases, the performances of the group-robust baselines are harmed much more severely. We interpreted that the reason for the degradation of these baselines is that they focus on the large loss samples, which are likely to be the noisy labels, as discussed in Sec 4. The noise label robust loss, GCE, improves the ERM with noisy labels, but its group robustness is insufficient. In addition, in the ablation study in Appendix C.1 shows that **(1)** utilizing entropy is the major contribution of the group and the noisy label robustness, as Theorem 1 stated, and **(2)** cooperation between the noisy-robust loss and the overconfident regularizer has an important role in its performance.

Additionally, we qualitatively show that our identification model can identify the SCF samples while being robust to the noisy labels. Concretely, we visualize the 2-D projection of the latent features (before the last linear layer) of the identification model on the Waterbirds with 30% label noise. This experiment has two implications. Firstly, the SCF set identified by our framework corresponds with the minority group (the true SCF samples). Specifically, the first row of Figure 4 (END) shows that both the minority group (red and green dots in the third column) and the SCF set are mainly

located around the middle of the images. Quantitatively, up to 30% of the SCF set consists of the minority group, which is higher than the actual proportion (5%). Secondly, the contamination of the SCF set with the noisy labeled samples is significantly mitigated by our framework. The first row of Figure 4 (END) shows that (1) the noise labels are almost identically distributed over the space, and (2) the noise labels do not severely overlap with the SCF set. This result substantiates that the proposed identification model (Sec 5.1) effectively identify the SCF samples while including less noisy samples. In contrast, the baseline identification model (the second row) shows an overlap between the noisy samples and the SCF set. This is in line with our claim: to utilize the uncertainty with the neural networks, the model should not memorize the noise labels and should have reliable uncertainty.

## 6.3 REAL-WORLD REGRESSION DATASET

In this subsection, we conduct experiments on the regression datasets with non-synthetic label noise to demonstrate the followings: **(1)** the END framework can be extended to a regression problem; **(2)** END achieve the group robustness under the non-artificial label noise. Particularly, we evaluate models on two drug-target affinity (DTA) benchmarks, Davis (Davis et al., 2011) and KIBA (Tang et al., 2014), see Appendix B for the details. Inputs of these datasets are drug molecules and protein sequences. The target value is a physical experiment-derived affinity values between the drug and the protein. We use the DeepDTA architecture (Öztürk et al., 2018) as the base architecture, see Appendix B for the details. Similar to the classification benchmarks, the DTA benchmarks have two characteristics. Firstly, the dataset has the spurious correlation: the DTA model typically relies on a single modality (e.g., predict its affinity by only leveraging the drug molecule, not considering the interaction with the target protein, or vice versa), which is inconsistent with the physicochemical laws (Özçelik et al., 2021; Yang et al., 2022). To get the worst group information of each benchmarks, we group data samples by their distinct drug molecules and target proteins, respectively. Secondly, the target values are naturally noisy due to the different environments of data acquisition (Davis et al., 2011; Tang et al., 2014; 2018). Thus, it can be seen as the non-synthetic label noise.

Since Lff and JTT cannot be extended to the regression problems, we propose an alternative baseline, "*hard*." Akin to JTT and END, the *hard* algorithm picks up the top-$K$ largest loss samples after the first phase of training. Next, another model is trained with the over-sampled training dataset with those hard samples. Table 2 shows that END outperforms others in terms of worst-group MSE metrics. We posit that it is the well-identified SCF set via the proposed uncertainty-based approach that contributes to this improvement. On the other hand, *hard* shows no improvement over ERM due to the oversampled noise labels.

Table 2: Evaluation results on the DTA datasets.

| Davis | MSE | WG MSE |
|---|---|---|
| ERM | 0.268 (0.01) | 0.860 (0.14) |
| Hard | 0.299 (0.02) | 1.196 (0.24) |
| END | 0.262 (0.01) | **0.785 (0.07)** |
| KIBA | MSE | WG MSE |
| ERM | 0.217 (0.04) | 8.915 (0.33) |
| Hard | 0.203 (0.02) | 8.748 (0.39) |
| END | 0.207 (0.05) | **8.358 (0.53)** |

## 7 DISCUSSION

In this study, we present a new approach that can significantly improve the group robustness under the label noise scenario. We theoretically show that the predictive uncertainty is the proper criterion for identifying the SCF samples. Upon this foundation, we propose the END framework consisting of two procedures. **(1)** Obtaining the SCF set via predictive uncertainty of the noise-robust model with reliable uncertainty; **(2)** Training the debiased model on the oversampled training set with the selected SCF samples. In practice, we empirically demonstrate that END achieves both the group and the noisy label robustness.

For future works, we discuss several potential areas of improvements. Firstly, the END framework adopts simple approaches (the MAE loss and the SGLD) for the identification model. Thus, the future works can employ a more advanced approach for the identification model which **(1)** obtains the reliable uncertainty and **(2)** prevents the memorization of the noisy label. Secondly, we only consider the *total* predictive uncertainty of the model in this study. However, the predictive uncertainty can be decomposed into two different types of uncertainty: *aleatoric* (uncertainty arising from the data *noise*) and *epistemic* (uncertainty arising from the model parameters) Kendall & Gal (2017); Oh & Shin (2022). If we disregard the decomposed aleatoric uncertainty, we believe that it could improve the END framework.

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

## APPENDIX

## A    FORMAL THEOREMS AND PROOFS

In this section, we formally state and prove our main theorem and an additional theorem.

- In A.1, we briefly summarize our conventions on notations.
- In A.2, we formalize the concept of "risk minimization" on our toy dataset, and provide the solution.
- In A.3, we list some basic definitions and lemmas that will be used in our theorem statements and proofs.
- In A.4, we re-state our main theorem formally.
- In A.5, we prove our main theorem.
- In A.6, we state the additional theorem.
- In A.7, we prove the additional theorem.

### A.1    NOTATIONS

Vectors are written in bold, while (one-dimensional) numbers are not. Random variables are always written in upper case, while plain values are written in lower case except for constants. For example, $\mathbf{X}$ is a random vector, while $w$ is a plain real number. Note that this is a slightly different from the notations in the main paper, where random variables are written in lower case.

$X \sim \mathcal{D}$ reads as "$X$ follows the distribution $\mathcal{D}$" or "$X$ is sampled from $\mathcal{D}$". $\mathbb{P}$ stands for the probability, and $\mathbb{E}$ stands for the expectation.

### A.2    RISK MINIMIZATION ON THE TOY DATASET

**Definition 1** (Risk-minimizing linear solution of a distribution). *Let $n \in \mathbb{N}$, and let $\mathcal{D}$ be a distribution on $\mathbb{R}^{n+1}$. We call $\boldsymbol{\beta}^* = (\beta_0, \beta_1, \cdots, \beta_n) \in \mathbb{R}^{n+1}$ the risk-minimizing linear solution of $\mathcal{D}$ if it is the solution of the minimization problem*

$$\boldsymbol{\beta}^* = \operatorname*{arg\,min}_{\beta_0, \beta_1, \cdots, \beta_n} \mathbb{E}_{(X_1, \cdots, X_n, Y) \sim \mathcal{D}} (\beta_0 + \sum_{i=1}^{n} \beta_i X_i - Y)^2 \tag{9}$$

**Definition 2** ($\mathcal{B}_\mathbf{p}$). *Let $\mathbf{p} = (p_1, \cdots, p_n) \in [0,1]^n$. We write $\mathcal{B}_\mathbf{p}$ for the joint distribution of $(X_1, \cdots, X_n)$ for independent random variables $X_i \in \{-1, 1\}$ where $X_i = 1$ with probability $p_i$. When $n = 1$, we simply write $\mathcal{B}_{p_1} = \mathcal{B}_{(p_1)}$*

**Definition 3** (The toy dataset). *Let $\mathbf{p} = (p_1, \cdots, p_n), \mathbf{p}' = (p'_1, \cdots, p'_n) \in [0,1]^n$ and $0 \leq \eta \leq 1$. We define a distribution $\mathcal{D} = \mathcal{D}^\eta_{\mathbf{p}, \mathbf{p}'}$ on $\mathbb{R}^{n+2}$ by its sampling procedure:*

1. *Sample $Z \sim \mathcal{B}_{0.5}$ uniformly.*

2. *If $Z = 1$, sample $\mathbf{X} = (X_1 \cdots, X_n) \sim \mathcal{B}_\mathbf{p}$. Otherwise if $Z = -1$, sample $\mathbf{X} = (X_1 \cdots, X_n) \sim \mathcal{B}_{\mathbf{p}'}$.*

3. *With probability $1 - \eta$, let $Y = Z$. Otherwise, let $Y = -Z$.*

4. *Output $(X_1, \cdots, X_n, Y, Z)$*

*Note that the mathematical objects used here can be interpreted as follows:*

- *$X_i$: $i$-th feature of the sample.*

- *$Z$: The true label of the sample, either positive(1) or negative($-1$).*

- *$p_i$: The probability that the $i$-th feature has value 1 for a positive sample.*

- *$p_i'$: The probability that the $i$-th feature has value 1 for a negative sample.*

- *$\eta$: The probability of "label noise".*

- *$Y$: The post-noise label of the sample*

**Proposition 1.** *Given $\mathbf{p} = (p_1, \cdots, p_n), \mathbf{p}' = (p_1', \cdots, p_n') \in [0, 1]^n$ and $0 \le \eta \le 1$, let $\boldsymbol{\beta}^* = (\beta_0, \beta_1, \cdots, \beta_n)$ be the solution of $\mathcal{D}_{\mathbf{p}, \mathbf{p}'}^\eta$. Then we have, for some $k = k(\mathbf{p}, \mathbf{p}') > 0$,*

$$\beta_i = k(1 - 2\eta) \frac{p_i - p_i'}{1 - \frac{1}{2}(2p_i - 1)^2 - \frac{1}{2}(2p_i' - 1)^2} \quad (i = 1, \cdots, n) \tag{10}$$

*and*

$$\beta_0 + \sum_{i=1}^n (p_i + p_i' - 1)\beta_i = 0 \tag{11}$$

*Proof.* All the expectations and probabilities that follow are with respect to $(\mathbf{X}, Y, Z) \sim \mathcal{D}_{\mathbf{p}, \mathbf{p}'}^\eta$. Let

$$\boldsymbol{\beta}^* = \arg\min_{\boldsymbol{\beta}} L(\boldsymbol{\beta}) = \arg\min_{\boldsymbol{\beta}} \mathbb{E}(\beta_0 + \sum_{i=1}^n \beta_i X_i - Y)^2$$

Using

- $\mathbb{E}[X_i] = \frac{1}{2}\mathbb{E}[X_i | Z = 1] + \frac{1}{2}\mathbb{E}[X_i | Z = -1] = p_i + p_i' - 1$

- $\mathbb{E}[X_i | Y = 1] = (1 - \eta)\mathbb{E}[X_i | Y = 1, Z = 1] + \eta\mathbb{E}[X_i | Y = 1, Z = -1] = (1 - \eta)(p_i - (1 - p_i)) + \eta(p_i' - (1 - p_i')) = 2p_i(1 - \eta) + 2p_i'\eta - 1$

- $\mathbb{E}[X_i | Y = -1] = (1 - \eta)\mathbb{E}[X_i | Y = -1, Z = -1] + \eta\mathbb{E}[X_i | Y = -1, Z = 1] = (1 - \eta)(p_i' - (1 - p_i')) + \eta(p_i - (1 - p_i)) = 2p_i\eta + 2p_i'(1 - \eta) - 1$

- For $i \ne j$, $\mathbb{E}[X_i X_j] = \frac{1}{2}(\mathbb{E}[X_i X_j | Z = 1] + \mathbb{E}[X_i X_j | Z = -1]) = \frac{1}{2}(\mathbb{E}[X_i | Z = 1]\mathbb{E}[X_j | Z = 1] + \mathbb{E}[X_i | Z = -1]\mathbb{E}[X_j | Z = -1]) = \frac{1}{2}((2p_i - 1)(2p_j - 1) + (2p_i' - 1)(2p_j' - 1))$

- $\mathbb{P}[Y = 1] = \mathbb{P}[Y = -1] = 1/2, \mathbb{E}[Y] = 0$

- $\mathbb{E}[X_i^2] = 1$

, we get

$$0 = \frac{1}{2} \frac{\partial}{\partial \beta_0} L(\boldsymbol{\beta}) \Big|_{\boldsymbol{\beta} = \boldsymbol{\beta}^*}$$

$$= \mathbb{E}(\beta_0 + \sum_{i=1}^n \beta_i X_i - Y) = \beta_0 + \sum_{i=1}^n \beta_i \mathbb{E}[X_i]$$

$$= \beta_0 + \sum_{i=1}^n (p_i + p_i' - 1)\beta_i$$

,

$$\beta_0 + \sum_{i=1}^{n} (p_i + p_i' - 1)\beta_i = 0 \tag{12}$$

(which proves equation 11) and

$$
\begin{aligned}
0 &= \frac{1}{2} \frac{\partial}{\partial \beta_i} L(\boldsymbol{\beta}) \Big|_{\boldsymbol{\beta}=\boldsymbol{\beta}^*} \\
&= \mathbb{E} X_i (\beta_0 + \sum_{j=1}^{n} \beta_j X_j - Y) \\
&= \beta_0 \mathbb{E} X_i + \beta_i \mathbb{E} X_i^2 + \sum_{j \neq i} \beta_j \mathbb{E}[X_i X_j] - \mathbb{E}[X_i Y] \\
&= \beta_0 \mathbb{E} X_i + \beta_i \mathbb{E} X_i^2 + \sum_{j \neq i} \beta_j \mathbb{E}[X_i X_j] - \frac{1}{2}\mathbb{E}[X_i | Y = 1] + \frac{1}{2}\mathbb{E}[X_i | Y = -1] \\
&= \beta_0 (p_i + p_i' - 1) + \beta_i + \frac{1}{2} \sum_{j \neq i} \beta_j ((2p_i - 1)(2p_j - 1) + (2p_i' - 1)(2p_j' - 1)) \\
&\quad - (1 - 2\eta)(p_i - p_i')
\end{aligned}
$$

,

$$\beta_0(p_i + p_i' - 1) + \beta_i + \frac{1}{2}\sum_{j \neq i} \beta_j ((2p_i - 1)(2p_j - 1) + (2p_i' - 1)(2p_j' - 1)) = (1 - 2\eta)(p_i - p_i') \tag{13}$$

Letting $w = \sum_{i=1}^{n} (2p_i - 1)\beta_i$ and $w' = \sum_{i=1}^{n} (2p_i' - 1)\beta_i$, we get

$$\beta_0 = -\frac{w + w'}{2} \tag{14}$$

from equation 12 and

$$\beta_0(p_i + p_i' - 1) + (p_i - \frac{1}{2})w + (p_i' - \frac{1}{2})w' + (1 - \frac{1}{2}(2p_i - 1)^2 - \frac{1}{2}(2p_i' - 1)^2)\beta_i = (1 - 2\eta)(p_i - p_i') \tag{15}$$

from equation 13. Plugging equation 14 into equation 15, we get

$$\frac{w - w'}{2}p_i - \frac{w - w'}{2}p_i' + (1 - \frac{1}{2}(2p_i - 1)^2 - \frac{1}{2}(2p_i' - 1)^2)\beta_i = (1 - 2\eta)(p_i - p_i')$$

hence

$$\beta_i = k' \frac{p_i - p_i'}{1 - \frac{1}{2}(2p_i - 1)^2 - \frac{1}{2}(2p_i' - 1)^2} \tag{16}$$

, where

$$k' = 1 - 2\eta - \frac{w - w'}{2}. \tag{17}$$

Plugging equation 16 back into equation 17 using the definitions of $w$ and $w'$, we get

$$k' = \frac{1 - 2\eta}{1 + \sum_{i=1}^{n} \frac{(p_i - p_i')^2}{1 - \frac{1}{2}(2p_i - 1)^2 - \frac{1}{2}(2p_i' - 1)^2}}$$

, so we conclude

$$\beta_i = k(1 - 2\eta) \frac{p_i - p_i'}{1 - \frac{1}{2}(2p_i - 1)^2 - \frac{1}{2}(2p_i' - 1)^2}$$

where

$$k = k(\mathbf{p}, \mathbf{p}') = \frac{1}{1 + \sum_{i=1}^{n} \frac{(p_i - p_i')^2}{1 - \frac{1}{2}(2p_i - 1)^2 - \frac{1}{2}(2p_i' - 1)^2}} > 0.$$

$\square$

### A.3 BASIC LEMMAS

**Definition 4** (cumulative distribution function $F$). *Let $\mathcal{D}$ be a distribution, and let $f$ be a real function on the domain of $\mathcal{D}$. Then for $y \in \mathbb{R}$, we define*

$$F_{f(\mathcal{D})}(y) = \mathbb{P}_{\mathbf{X} \in \mathcal{D}}[f(\mathbf{X}) \leq y]$$

*We treat "$F(f(\mathbf{X}))$" as a shorthand of $F_{f(\mathcal{D})}(f(\mathbf{X}))$ when the definition of $\mathcal{D}$ is clear from the context.*

**Definition 5** (anti-concentration of a discrete random variable). *Let $W$ be a discrete random variable. We write*

$$AC(W) = \max_w \mathbb{P}[W = w]$$

We make use of the following estimates of $\mathbb{P}[F(W) \leq a)]$ and $\mathbb{P}[F(W) \geq 1 - a]$ for discrete random variables $W$ throughout our proof:

**Lemma 1.** *For any discrete random variable $W$ and any $a \in [0, 1]$, we have*

$$a - AC(W) < \mathbb{P}[F(W) \leq a] \leq a \tag{18}$$

*and*

$$a - AC(W) < \mathbb{P}[F(W) \geq 1 - a] \leq a \tag{19}$$

*Proof.* It is enough to prove equation 18. equation 19 can be shown from equation 18 by letting $W = -W$. Let

$$w^* = \max \{w \in \mathbb{R} | F(w) \leq a\}$$

We have

$$\mathbb{P}[F(W) \leq a] = \mathbb{P}[W \leq w^*] = F(w^*)$$

Since

$$F(w^*) \leq a$$

by the definition of $w^*$, we have the inequality on the right hand side. To show the inequality on the left hand side, suppose $F(w^*) \leq a - AC(W)$. Let $w^{**}$ be the smallest real number in the range of $W$ bigger than $w^*$. By the definition of $w^*$, we have $F(w^{**}) > a$. Then,

$$AC(W) \geq \mathbb{P}[W = w^{**}] = \mathbb{P}[W \leq w^{**}] - \mathbb{P}[W \leq w^*] = F(w^{**}) - F(w^*) > a - (a - AC(W)) = AC(W)$$

, which is a contradiction.

$\square$

Upper bounds on $AC(W)$ are called *anti-concentration inequalities* in the literature (Krishnapur (2016)). One such inequality is *Littlewood-Offord inequality*, and a version of it that encompasses our case was proved in Juškevičius & Kurauskas (2019):

**Lemma 2.** *Let $W = \sum_{i=1}^n \beta_i X_i$, where $\beta_i \neq 0$ and $\mathbf{X} = (X_1, \cdots, X_n) \sim \mathcal{B}_{\mathbf{p}}$ for $\mathbf{p} \in [\epsilon, 1 - \epsilon]^n$. Then we have*

$$AC(W) \leq \frac{C}{\sqrt{n}}$$

*for some $C = C(\epsilon) > 0$.*

*Proof.* This is a direct consequence of Corollary 2 in Juškevičius & Kurauskas (2019). $\square$

An application of Lemma 1 is the following.

**Lemma 3.** *Let $W$ be a discrete random variable. Let $\alpha = \mathbb{P}[W < 0]$ and $\gamma = AC(W)$. For any $\delta > 0$, we have*

$$\mathbb{P}[F(|W|) \leq \delta \text{ and } F(W) > \delta] \leq 2\alpha + \gamma$$

*and*

$$\mathbb{P}[F(W) \leq \delta \text{ and } F(|W|) > \delta] \leq \alpha$$

*In particular, for any event $E$, for any $\delta > 2\gamma$, we have*

$$\mathbb{P}[E \mid F(|W|) \leq \delta] \geq \mathbb{P}[E \mid F(W) \leq \delta] \cdot \frac{1}{1 + (2\alpha + \gamma)/(\delta - \gamma)} - \frac{\alpha}{\delta - \gamma}$$

*Proof.* We have

$$
\begin{aligned}
\mathbb{P}[F(|W|) \leq \delta \text{ and } F(W) > \delta] &\leq \mathbb{P}[\delta < F(W) \leq \delta + \alpha] + \mathbb{P}[W < 0] \\
&\quad + \mathbb{P}[W \geq 0 \text{ and } F(|W|) \leq \delta \text{ and } F(W) > \delta + \alpha] \\
&\leq 2\alpha + \gamma + \mathbb{P}[W \geq 0 \text{ and } F(|W|) \leq \delta \text{ and } F(W) > \delta + \alpha] \\
&\quad (\text{by Lemma 1}) \\
&= 2\alpha + \gamma
\end{aligned}
$$

, where the last equality is valid because if $W \geq 0$, then $F(|W|) \leq \delta$ and $F(|W|) > \delta + \alpha$ cannot hold simultaneously since if they do, $\delta + \alpha < F(W) = \mathbb{P}[W' \leq W] = \mathbb{P}[W' \leq |W|] \leq \mathbb{P}[W' < 0] + \mathbb{P}[|W'| < |W|] = \alpha + F(|W|) \leq \delta + \alpha$, which is a contradiction.

Similarly, we have

$$
\begin{aligned}
\mathbb{P}[F(W) \leq \delta \text{ and } F(|W|) > \delta] &\leq \mathbb{P}[W < 0] + \mathbb{P}[W \geq 0 \text{ and } F(W) \leq \delta \text{ and } F(|W|) > \delta] \\
&= \alpha + \mathbb{P}[W \geq 0 \text{ and } F(W) \leq \delta \text{ and } F(|W|) > \delta] \\
&= \alpha
\end{aligned}
$$

We can show the remaining statement as follows:

$$
\begin{aligned}
\mathbb{P}[E \mid F(|W|) \leq \delta] &= \frac{\mathbb{P}[E \text{ and } F(|W|) \leq \delta]}{\mathbb{P}[F(|W|) \leq \delta]} \\
&\geq \frac{\mathbb{P}[E \text{ and } F(W) \leq \delta] - \mathbb{P}[F(W) \leq \delta \text{ and } F(|W|) > \delta]}{\mathbb{P}[F(|W|) \leq \delta]} \\
&\geq \frac{\mathbb{P}[E \text{ and } F(W) \leq \delta]}{\mathbb{P}[F(|W|) \leq \delta]} - \frac{\alpha}{\delta - \gamma} \\
&\geq \frac{\mathbb{P}[E \text{ and } F(W) \leq \delta]}{\mathbb{P}[F(W) \leq \delta] + \mathbb{P}[F(|W|) \leq \delta \text{ and } F(W) > \delta]} - \frac{\alpha}{\delta - \gamma} \\
&\geq \frac{\mathbb{P}[E \text{ and } F(W) \leq \delta]}{\mathbb{P}[F(W) \leq \delta] + 2\alpha + \gamma} - \frac{\alpha}{\delta - \gamma} \\
&= \frac{\mathbb{P}[E \text{ and } F(W) \leq \delta]}{\mathbb{P}[F(W) \leq \delta](1 + (2\alpha + \gamma)/\mathbb{P}[F(W) \leq \delta])} - \frac{\alpha}{\delta - \gamma} \\
&\geq \mathbb{P}[E \mid F(W) \leq \delta] \cdot \frac{1}{1 + (2\alpha + \gamma)/(\delta - \gamma)} - \frac{\alpha}{\delta - \gamma}
\end{aligned}
$$

$\square$

Also, we need a quantitative version of multi-dimensional central limit theorem:

**Lemma 4** (Bentkus (2005)). *Let $\mathbf{V}_i \in \mathbb{R}^d$ ($i = 1, \cdots, n$) be independent random variables with $\mathbb{E}[\mathbf{V}_i] = 0$. Let*

$$
\mathbf{S} = \sum_{i=1}^{n} \mathbf{V}_i.
$$

*If the covariance matrix $\Sigma = \Sigma(\mathbf{S})$ is invertible, for $\mathbf{Z} \sim N(0, \Sigma)$, we have*

$$
|\mathbb{P}[\mathbf{S} \in U] - \mathbb{P}[\mathbf{Z} \in U]| \leq C\gamma \text{ for any convex } U \subseteq \mathbb{R}^2
$$

*, where $C$ is an absolute constant and*

$$
\gamma = \sum_{i=1}^{n} \mathbb{E}[\|\Sigma^{-1}\mathbf{V}_i\|_2^3].
$$

## A.4 THE ENTROPY THEOREM

**Definition 6** (entropy function). *For $\boldsymbol{\beta} = (\beta_0, \cdots, \beta_n) \in \mathbb{R}^{n+1}$, we define the entropy function $H_{\boldsymbol{\beta}} : \mathbb{R}^n \to \mathbb{R}$ as follows:*

$$H_{\boldsymbol{\beta}}(x_1, \cdots, x_n) = H\big(\frac{\exp(\beta_0 + \sum_{i=1}^n \beta_i x_i)}{1 + \exp(\beta_0 + \sum_{i=1}^n \beta_i x_i)}\big), \quad \text{where } H(p) = -p \log p - (1-p) \log(1-p)$$

*Note that $H_{\boldsymbol{\beta}}(x_1, \cdots, x_n)$ can be expressed as $f(|\beta_0 + \sum_{i=1}^n \beta_i x_i|)$ for some monotonically decreasing function $f$.*

**Definition 7** (sign-respecting function). *Let $A_+ = \{(p, p') \in [0,1] \times [0,1] : p > p'\}$ and $A_- = \{(p, p') \in [0,1] \times [0,1] : p < p'\}$. A function $\Lambda : ([0,1] \times [0,1]) \backslash \{(x,x) | x \in [0,1]\} = A_+ \cup A_- \to \mathbb{R}$ will be called sign-respecting if it is continuous on each of $A_+$ and $A_-$, positive on $A_+$ and negative on $A_-$. For example, $f(p, p') = \frac{p-p'}{|p-p'|}$ is sign-respecting.*

**Definition 8** (spurious cue score function). *Given a sign-respecting function $\Lambda$, under the distribution defined in Definition 3, we define the spurious cue score function $\Psi_{\mathbf{p}, \mathbf{p}'}^{\Lambda} : \mathbb{R}^n \to \mathbb{R}$ as follows:*

$$\Psi_{\mathbf{p}, \mathbf{p}'}^{\Lambda}(\mathbf{X}) = \sum_{i=1}^n \Lambda(p_i, p_i') X_i$$

*Roughly speaking, $\Psi_{\mathbf{p}, \mathbf{p}'}^{\Lambda}$ measures how easy a given positive sample $\mathbf{X}$ is in terms of the number of label-compatible features. Here, a feature $X_i$ of a positive sample is label-compatible if either $X_i = 1$ and $p_i > p_i'$ (i.e. positive samples have higher probability of having $X_i = 1$) or $X_i = -1$ and $p_i' > p_i$ (i.e. negative samples have higher probability of having $X_i = 1$). Here, the "number" is weighted in terms of $\Lambda(p_i, p_i')$.*

*If we want to measure a similar thing for a negative sample, we can use $\Psi_{\mathbf{p}', \mathbf{p}}$ instead.*

**Theorem 1** (formal). *Let $\Lambda$ be a sign-respecting function. Given $0 < \epsilon < 1/2$, there exists $\delta_0 = \delta_0(\Lambda, \epsilon) > 0$ such that for any $0 < \delta \le \delta_0$, for sufficiently large $n$ ($n \ge N$ for some $N = N(\Lambda, \epsilon, \delta)$), the following holds:*

*For any combination of*

- $0 \le \eta < 1/2$

- $\mathbf{p} = (p_1, \cdots, p_n), \mathbf{p}' = (p_1', \cdots, p_n') \in [\epsilon, 1-\epsilon]^n$ with $|p_i - p_i'| \ge \epsilon$ $(1 \le i \le n)$

*, when $\boldsymbol{\beta}^* = (\beta_0, \beta_1 \cdots, \beta_n)$ is the risk-minimizing linear solution of $(\mathbf{X}, Y)$ for $(\mathbf{X}, Y, Z) \sim \mathcal{D}_{\mathbf{p}, \mathbf{p}'}^{\eta}$, we have*

$$\mathbb{P}_{\mathbf{X}, Y, Z \sim \mathcal{D}_{\mathbf{p}, \mathbf{p}'}^{\eta}} [\ R_{\mathbf{p}, \mathbf{p}'}^{\Lambda}(\mathbf{X}, Z) \le \epsilon \ | \ F(H_{\boldsymbol{\beta}^*}(\mathbf{X})) \ge 1 - \delta \ ] \ge 1 - \epsilon. \tag{20}$$

*, where $R_{\mathbf{p}, \mathbf{p}'}^{\Lambda}$ is the spurious cue score rank function defined as*

$$R_{\mathbf{p}, \mathbf{p}'}^{\Lambda}(\mathbf{X}, Z) = \mathbf{1}_{Z=1} F_{\Psi_{\mathbf{p}, \mathbf{p}'}(\mathcal{B}_{\mathbf{p}})}(\Psi_{\mathbf{p}, \mathbf{p}'}^{\Lambda}(\mathbf{X})) + \mathbf{1}_{Z=-1} F_{\Psi_{\mathbf{p}', \mathbf{p}}^{\Lambda}(\mathcal{B}_{\mathbf{p}'})}(\Psi_{\mathbf{p}', \mathbf{p}}^{\Lambda}(\mathbf{X}))$$

*That is, $R_{\mathbf{p}, \mathbf{p}'}^{\Lambda}(\mathbf{X}, Z)$ is the rank of the spurious cue score of $\mathbf{X}$ within its ground truth label $Z$.*

*This means intuitively that under our dataset generation process (as described by Definition 3) with some minor conditions (Those for $\mathbf{p}$ and $\mathbf{p}'$), relatively high (top-$\delta$) entropy w.r.t. $\boldsymbol{\beta}^*$ implies relatively small (bottom-$\epsilon$ within the ground truth label $Z$) spurious cue score with high probability (probability at least $1 - \epsilon$).*

## A.5 PROOF OF THEOREM 1

### A.5.1 A REDUCTION

We reduce Theorem 1 to the following variant:

**Lemma 5.** *Let $\Lambda$ be a sign-respecting function. Given $0 < \epsilon < 1/2$, there exists $\delta_0 = \delta_0(\Lambda, \epsilon) > 0$ such that for any $0 < \delta_{min} \leq \delta_0$, for sufficiently large $n$ ($n \geq N$ for some $N = N(\Lambda, \epsilon, \delta_{min})$), the following holds:*

*For any combination of*

- *$0 \leq \eta < 1/2$*

- *$\mathbf{p} = (p_1, \cdots, p_n), \mathbf{p}' = (p_1', \cdots, p_n') \in [\epsilon, 1 - \epsilon]^n$ with $|p_i - p_i'| \geq \epsilon$ ($1 \leq i \leq n$)*

- *$\delta \in [\delta_{min}, \delta_0]$*

*, when $\boldsymbol{\beta}^* = (\beta_0, \beta_1 \cdots, \beta_n)$ is the risk-minimizing linear solution of $(\mathbf{X}, Y)$ for $(\mathbf{X}, Y, Z) \sim \mathcal{D}^\eta_{\mathbf{p},\mathbf{p}''}$,*

$$\mathbb{P}_{\mathbf{X} \sim \mathcal{B}_{\mathbf{p}}}[\ F(\Psi^\Lambda_{\mathbf{p},\mathbf{p}'}(\mathbf{X})) \leq \epsilon \ | \ F(H_{\boldsymbol{\beta}^*}(\mathbf{X})) \geq 1 - \delta\ ] \geq 1 - \epsilon. \tag{21}$$

*and*

$$\mathbb{P}_{\mathbf{X} \sim \mathcal{B}_{\mathbf{p}'}}[\ F(\Psi^\Lambda_{\mathbf{p}',\mathbf{p}}(\mathbf{X})) \leq \epsilon \ | \ F(H_{\boldsymbol{\beta}^*}(\mathbf{X})) \geq 1 - \delta\ ] \geq 1 - \epsilon. \tag{22}$$

*where $H_{\boldsymbol{\beta}^*}$ is the entropy function (Definition 6), $\Psi^\Lambda_{\mathbf{p},\mathbf{p}'}$ and $\Psi^\Lambda_{\mathbf{p}',\mathbf{p}}$ are the spurious cue score functions (Definition 8) and $F$ are the cumulative distribution functions(Definition 4) related to them with respect to $B_{\mathbf{p}}$ or $B_{\mathbf{p}'}$.*

*The key differences from Theorem 1 are (1) the introduction of $\delta_{min}$ and (2) that equation 20 has been separated into equation 21 and equation 22.*

Before presenting the proof of Lemma 5, we will show how to derive Theorem 1 from Lemma 5.

*proof of Theorem 1 assuming Lemma 5.* Let $\Lambda$ be a sign-respecting function, and let $0 < \epsilon < 1/2$. Let

$$\delta_0^* = \delta_0^{Lemma5}(\Lambda, \frac{\epsilon}{2})$$

and take

$$\delta_0 = \frac{\delta_0^*}{3}$$

. Given $\delta$ with

$$0 < \delta < \delta_0$$

, let

$$\delta_{min} = \frac{\delta\epsilon}{2}$$

and take

$$N = \max(N^{Lemma5}(\Lambda, \epsilon', \delta_{min}), (\frac{6C}{\delta_0^*})^2, (\frac{4C}{\delta})^2)$$

, where $C = C(\epsilon)$ is from Lemma 2. To show that $N$ satisfies the theorem statement, let $n \geq N$. Since $\delta_{min} \leq \delta_0^*$, by Lemma 5, we have

$$\forall \delta_{min} \leq \delta' \leq \delta_0^*, \quad \mathbb{P}_{\mathbf{X} \sim \mathcal{B}_{\mathbf{p}}}[\ F(\Psi^\Lambda_{\mathbf{p},\mathbf{p}'}(\mathbf{X})) \leq \frac{\epsilon}{2} \ | \ F(H_{\boldsymbol{\beta}^*}(\mathbf{X})) \geq 1 - \delta'\ ] \geq 1 - \frac{\epsilon}{2} \tag{23}$$

and

$$\forall \delta_{min} \leq \delta' \leq \delta_0^*, \quad \mathbb{P}_{\mathbf{X} \sim \mathcal{B}_{\mathbf{p}'}}[\ F(\Psi^\Lambda_{\mathbf{p}',\mathbf{p}}(\mathbf{X})) \leq \frac{\epsilon}{2} \ | \ F(H_{\boldsymbol{\beta}^*}(\mathbf{X})) \geq 1 - \delta'\ ] \geq 1 - \frac{\epsilon}{2}. \tag{24}$$

Our goal is to show that

$$\text{Goal: } \mathbb{P}_{\mathbf{X},Y,Z \sim \mathcal{D}^\eta_{\mathbf{p},\mathbf{p}'}}[\ R^\Lambda_{\mathbf{p},\mathbf{p}'}(\mathbf{X}, Z)) \le \epsilon \ | \ F(H_{\boldsymbol{\beta}^*}(\mathbf{X})) \ge 1 - \delta \ ] \ge 1 - \epsilon \tag{25}$$

, where

$$R^\Lambda_{\mathbf{p},\mathbf{p}'}(\mathbf{X}, Z) = \mathbf{1}_{Z=1} F_{\Psi_{\mathbf{p},\mathbf{p}'}(\mathcal{B}_{\mathbf{p}})}(\Psi^\Lambda_{\mathbf{p},\mathbf{p}'}(\mathbf{X})) + \mathbf{1}_{Z=-1} F_{\Psi^\Lambda_{\mathbf{p}',\mathbf{p}}(\mathcal{B}_{\mathbf{p}'})}(\Psi^\Lambda_{\mathbf{p}',\mathbf{p}}(\mathbf{X})).$$

To simplify the equations that follow, let us make the following definitions:

- $a^* = \min\left\{a \in \mathbb{R} | F_{H_{\boldsymbol{\beta}^*}(\mathcal{D}^\eta_{\mathbf{p},\mathbf{p}'})}(a) \ge 1 - \delta)\right\}$

- $p = \mathbb{P}_{\mathbf{X},Y,Z \sim \mathcal{D}^\eta_{\mathbf{p},\mathbf{p}'}}[Z = 1 | H_{\boldsymbol{\beta}^*}(\mathbf{X}) \ge a^*]$

- $\delta_1 = 1 - F_{H_{\boldsymbol{\beta}^*}(\mathcal{B}_{\mathbf{p}})}(a^*)$

- $\delta_2 = 1 - F_{H_{\boldsymbol{\beta}^*}(\mathcal{B}_{\mathbf{p}'})}(a^*)$

We have the following (in)equalities:

1.
$$\mathbb{P}_{\mathbf{X} \sim \mathcal{B}_{\mathbf{p}}}[F(H_{\boldsymbol{\beta}^*}(\mathbf{X})) \ge 1 - \delta_1] = 2p\mathbb{P}_{\mathbf{X},Y,Z \sim \mathcal{D}^\eta_{\mathbf{p},\mathbf{p}'}}[F(H_{\boldsymbol{\beta}^*}(\mathbf{X})) \ge 1 - \delta] \tag{26}$$

and

$$\mathbb{P}_{\mathbf{X} \sim \mathcal{B}_{\mathbf{p}'}}[F(H_{\boldsymbol{\beta}^*}(\mathbf{X})) \ge 1 - \delta_2] = 2(1-p)\mathbb{P}_{\mathbf{X},Y,Z \sim \mathcal{D}^\eta_{\mathbf{p},\mathbf{p}'}}[F(H_{\boldsymbol{\beta}^*}(\mathbf{X})) \ge 1 - \delta] \tag{27}$$

*Proof.* We have

$$\begin{aligned}
\mathbb{P}_{\mathbf{X} \sim \mathcal{B}_{\mathbf{p}}}[F(H_{\boldsymbol{\beta}^*}(\mathbf{X})) \ge 1 - \delta_1] &= \mathbb{P}_{\mathbf{X} \sim \mathcal{B}_{\mathbf{p}}}[H_{\boldsymbol{\beta}^*}(\mathbf{X}) \ge a^*] \\
&= \mathbb{P}_{\mathbf{X},Y,Z \sim \mathcal{D}^\eta_{\mathbf{p},\mathbf{p}'}}[H_{\boldsymbol{\beta}^*}(\mathbf{X}) \ge a^* | Z = 1] \\
&= \frac{\mathbb{P}_{\mathbf{X},Y,Z \sim \mathcal{D}^\eta_{\mathbf{p},\mathbf{p}'}}[Z = 1 | H_{\boldsymbol{\beta}^*}(\mathbf{X}) \ge a^*] \cdot \mathbb{P}_{\mathbf{X},Y,Z \sim \mathcal{D}^\eta_{\mathbf{p},\mathbf{p}'}}[H_{\boldsymbol{\beta}^*}(\mathbf{X}) \ge a^*]}{\mathbb{P}_{\mathbf{X},Y,Z \sim \mathcal{D}^\eta_{\mathbf{p},\mathbf{p}'}}[Z = 1]} \\
&= 2p\mathbb{P}_{\mathbf{X},Y,Z \sim \mathcal{D}^\eta_{\mathbf{p},\mathbf{p}'}}[H_{\boldsymbol{\beta}^*}(\mathbf{X}) \ge a^*] \\
&= 2p\mathbb{P}_{\mathbf{X},Y,Z \sim \mathcal{D}^\eta_{\mathbf{p},\mathbf{p}'}}[F(H_{\boldsymbol{\beta}^*}(\mathbf{X})) \ge 1 - \delta]
\end{aligned}$$

, which proves equation 26. equation 27 can be proved similarly. $\qquad\square$

2.
$$\delta_1, \delta_2 \le \delta_0^* \tag{28}$$

*Proof.* We have

$$\begin{aligned}
\delta_1 - AC_{\mathbf{X} \sim \mathcal{B}_{\mathbf{p}}}(H_{\boldsymbol{\beta}^*}(\mathbf{X})) &\le \mathbb{P}_{\mathbf{X} \sim \mathcal{B}_{\mathbf{p}}}[F(H_{\boldsymbol{\beta}^*}(\mathbf{X})) \ge 1 - \delta_1] \text{ (By Lemma 1)} \\
&= 2p\mathbb{P}_{\mathbf{X},Y,Z \sim \mathcal{D}^\eta_{\mathbf{p},\mathbf{p}'}}[F(H_{\boldsymbol{\beta}^*}(\mathbf{X})) \ge 1 - \delta] \text{ (By equation 26)} \\
&\le 2p\delta \text{ (By Lemma 1)} \\
&\le 2\delta \le 2\delta_0 \le \frac{2}{3}\delta_0^*.
\end{aligned}$$

Since

$$AC_{\mathbf{X}\sim\mathcal{B}_{\mathbf{p}}}(H_{\boldsymbol{\beta}^*}(\mathbf{X})) = AC_{\mathbf{X}\sim\mathcal{B}_{\mathbf{p}}}\left(\left|\sum_{i=1}^{n}\boldsymbol{\beta}_i^* X_i\right|\right)$$

$$\leq 2AC_{\mathbf{X}\sim\mathcal{B}_{\mathbf{p}}}(\sum_{i=1}^{n}\boldsymbol{\beta}_i^* X_i)$$

$$\leq \frac{2C}{\sqrt{n}} \text{ (By Lemma 2)}$$

$$\leq \frac{\delta_0^*}{3} \text{ (Since } n \geq (\frac{6C}{\delta_0^*})^2)$$

, this proves $\delta_1 \leq \delta_0^*$. $\delta_2 \leq \delta_0^*$ can be proved similarly. $\qquad\square$

3.

$$\delta_1 \geq p\delta, \quad \delta_2 \geq (1-p)\delta \tag{29}$$

*Proof.* We have

$$\delta_1 \geq \mathbb{P}_{\mathbf{X}\sim\mathcal{B}_{\mathbf{p}}}[F(H_{\boldsymbol{\beta}^*}(\mathbf{X})) \geq 1 - \delta_1] \text{ (By Lemma 1)}$$

$$= 2p\mathbb{P}_{\mathbf{X},Y,Z\sim\mathcal{D}_{\mathbf{p},\mathbf{p}'}^{\eta}}[F(H_{\boldsymbol{\beta}^*}(\mathbf{X})) \geq 1 - \delta] \text{ (By equation 26)}$$

$$\geq 2p(\delta - AC_{\mathbf{X},Y,Z\sim\mathcal{D}_{\mathbf{p},\mathbf{p}'}^{\eta}}(H_{\boldsymbol{\beta}^*}(\mathbf{X}))) \text{ (By Lemma 1)}$$

$$\geq 2p(\delta - \frac{1}{2}(AC_{\mathbf{X}\sim\mathcal{B}_{\mathbf{p}}}(H_{\boldsymbol{\beta}^*}(\mathbf{X})) + AC_{\mathbf{X}\sim\mathcal{B}_{\mathbf{p}'}}(H_{\boldsymbol{\beta}^*}(\mathbf{X}))))$$

$$= 2p(\delta - \frac{1}{2}(AC_{\mathbf{X}\sim\mathcal{B}_{\mathbf{p}}}\left(\left|\sum_{i=1}^{n}\boldsymbol{\beta}_i^* X_i\right|\right) + AC_{\mathbf{X}\sim\mathcal{B}_{\mathbf{p}'}}\left(\left|\sum_{i=1}^{n}\boldsymbol{\beta}_i^* X_i\right|\right)))$$

$$\geq 2p(\delta - (AC_{\mathbf{X}\sim\mathcal{B}_{\mathbf{p}}}(\sum_{i=1}^{n}\boldsymbol{\beta}_i^* X_i) + AC_{\mathbf{X}\sim\mathcal{B}_{\mathbf{p}'}}(\sum_{i=1}^{n}\boldsymbol{\beta}_i^* X_i)))$$

$$\geq 2p(\delta - \frac{2C}{\sqrt{n}}) \text{ (By applying Lemma 2 twice)}$$

$$\geq p\delta \text{ (Since } n \geq (\frac{4C}{\delta})^2)$$

, which proves $\delta_1 \geq p\delta$. $\delta_2 \geq (1-p)\delta$ can be proved similarly. $\qquad\square$

Using these inequalities, we can show equation 25 as follows:

$$\mathbb{P}_{\mathbf{X},Y,Z\sim\mathcal{D}^\eta_{\mathbf{p},\mathbf{p}'}}[\ R^\Lambda_{\mathbf{p},\mathbf{p}'}(\mathbf{X},Z)) \leq \epsilon \mid F(H_{\boldsymbol{\beta}^*}(\mathbf{X})) \geq 1-\delta\ ]$$

$$\geq \mathbb{P}_{\mathbf{X},Y,Z\sim\mathcal{D}^\eta_{\mathbf{p},\mathbf{p}'}}[\ R^\Lambda_{\mathbf{p},\mathbf{p}'}(\mathbf{X},Z)) \leq \frac{\epsilon}{2} \mid F(H_{\boldsymbol{\beta}^*}(\mathbf{X})) \geq 1-\delta\ ]$$

$$= \mathbb{P}_{\mathbf{X},Y,Z\sim\mathcal{D}^\eta_{\mathbf{p},\mathbf{p}'}}[\ R^\Lambda_{\mathbf{p},\mathbf{p}'}(\mathbf{X},Z)) \leq \frac{\epsilon}{2} \mid H_{\boldsymbol{\beta}^*}(\mathbf{X}) \geq a^*\ ] \text{ (By definition of } a^*)$$

$$= p\cdot\mathbb{P}_{\mathbf{X},Y,Z\sim\mathcal{D}^\eta_{\mathbf{p},\mathbf{p}'}}[\ F_{\Psi_{\mathbf{p},\mathbf{p}'}(\mathcal{B}_{\mathbf{p}})}(\Psi_{\mathbf{p},\mathbf{p}'}(\mathbf{X})) \leq \frac{\epsilon}{2} \mid H_{\boldsymbol{\beta}^*}(\mathbf{X}) \geq a^*, Z=1\ ]$$

$$+ (1-p)\cdot\mathbb{P}_{\mathbf{X},Y,Z\sim\mathcal{D}^\eta_{\mathbf{p},\mathbf{p}'}}[\ F_{\Psi_{\mathbf{p}',\mathbf{p}}(\mathcal{B}_{\mathbf{p}'})}(\Psi_{\mathbf{p}',\mathbf{p}}(\mathbf{X})) \leq \frac{\epsilon}{2} \mid H_{\boldsymbol{\beta}^*}(\mathbf{X}) \geq a^*, Z=-1\ ]$$

$$= p\cdot\mathbb{P}_{\mathbf{X}\sim\mathcal{B}_{\mathbf{p}}}[\ F_{\Psi_{\mathbf{p},\mathbf{p}'}(\mathcal{B}_{\mathbf{p}})}(\Psi_{\mathbf{p},\mathbf{p}'}(\mathbf{X})) \leq \frac{\epsilon}{2} \mid H_{\boldsymbol{\beta}^*}(\mathbf{X}) \geq a^*\ ]$$

$$+ (1-p)\cdot\mathbb{P}_{\mathbf{X}\sim\mathcal{B}_{\mathbf{p}'}}[\ F_{\Psi_{\mathbf{p}',\mathbf{p}}(\mathcal{B}_{\mathbf{p}'})}(\Psi_{\mathbf{p}',\mathbf{p}}(\mathbf{X})) \leq \frac{\epsilon}{2} \mid H_{\boldsymbol{\beta}^*}(\mathbf{X}) \geq a^*\ ]$$

$$= p\cdot\mathbb{P}_{\mathbf{X}\sim\mathcal{B}_{\mathbf{p}}}[\ F_{\Psi_{\mathbf{p},\mathbf{p}'}(\mathcal{B}_{\mathbf{p}})}(\Psi_{\mathbf{p},\mathbf{p}'}(\mathbf{X})) \leq \frac{\epsilon}{2} \mid F(H_{\boldsymbol{\beta}^*}(\mathbf{X})) \geq 1-\delta_1\ ]$$

$$+ (1-p)\cdot\mathbb{P}_{\mathbf{X}\sim\mathcal{B}_{\mathbf{p}'}}[\ F_{\Psi_{\mathbf{p}',\mathbf{p}}(\mathcal{B}_{\mathbf{p}'})}(\Psi_{\mathbf{p}',\mathbf{p}}(\mathbf{X})) \leq \frac{\epsilon}{2} \mid F(H_{\boldsymbol{\beta}^*}(\mathbf{X})) \geq 1-\delta_2\ ]$$

$$\geq (1-\frac{\epsilon}{2})(p\mathbf{1}_{\delta_1\geq\delta_{min}} + (1-p)\mathbf{1}_{\delta_2\geq\delta_{min}}) \text{ (By equation 23, equation 24 and equation 28)}$$

$$\geq (1-\frac{\epsilon}{2})(p\mathbf{1}_{p\geq\delta_{min}/\delta} + (1-p)\mathbf{1}_{1-p\geq\delta_{min}/\delta}) \text{ (By equation 29)}$$

$$\geq (1-\frac{\epsilon}{2})(1-\delta_{min}/\delta) \text{ (Case analysis based on the magnitude of } p)$$

$$= (1-\frac{\epsilon}{2})^2$$

$$\geq 1-\epsilon$$

$\square$

### A.5.2 Proof of Lemma 5

Now, let us prove Lemma 5. The proof relies on the calculation results of Proposition 1 and the following lemma.

**Lemma 6.** *Given $0 < \epsilon < 1/2$ and $M \geq 1$, there exists $\delta_0 = \delta_0(\epsilon, M) > 0$ such that for any $0 < \delta_{min} \leq \delta_0$, for sufficiently large $n$ (i.e. $n \geq N$ for some $N = N(\epsilon, M, \delta_{\min})$), the following holds:*

*For any combination of*

- $\mathbf{p} = (p_1, \cdots, p_n) \in [\epsilon, 1-\epsilon]^n$

- $a_1, \cdots, a_n, b_1, \cdots, b_n \in [\frac{1}{M}, M]$

- $\delta \in [\delta_{min}, \delta_0]$

*, we have*

$$\mathbb{P}_{\mathbf{X}\sim\mathcal{B}_{\mathbf{p}}}[F(\sum_{i=1}^n a_i X_i) \leq \epsilon \mid F(\sum_{i=1}^n b_i X_i) \leq \delta] \geq 1-\epsilon \tag{30}$$

Before presenting the proof of this lemma, we will show how to conclude Lemma 5 from it.

Let $0 < \epsilon < 1/2$, $M \geq 1$. It is enough to find $\delta_0$ that satisfies the theorem statement where only equation 21 is considered but not equation 22. This is because since then a symmetric argument that accounts for equation 22 can be made and we can take the minimum of the both $\delta_0$.

Let

$$\beta(a,b) = \frac{a-b}{1 - \frac{1}{2}(2a-1)^2 - \frac{1}{2}(2b-1)^2}.$$

Take $M = M(\Lambda, \epsilon)$ such that

$$\frac{1}{M} \leq \min_{(a,b) \in I_\epsilon} |\beta(a,b)|, \min_{(a,b) \in I_\epsilon} |\Lambda(a,b)| \quad \text{and} \quad \max_{(a,b) \in I_\epsilon} |\beta(a,b)|, \max_{(a,b) \in I_\epsilon} |\Lambda(a,b)| \leq M$$

, where $I_\epsilon = \{(a,b) \in [\epsilon, 1-\epsilon] \times [\epsilon, 1-\epsilon] : |a-b| \geq \epsilon\}$.

Let $\delta_0 = \delta_0^{Lemma6}(\epsilon/2, M)$, where $\delta_0^{Lemma6}(\epsilon, M)$ stands for the $\delta_0$ found by using Lemma 6. Suppose $0 < \delta_{min} \leq \delta_0$. Suppose $\delta \in [\delta_{min}, \delta_0]$, $\mathbf{p}$, $\mathbf{p}'$ and $\eta$ have been chosen, and $\boldsymbol{\beta}^* = (\beta_0, \cdots, \beta_n)$ has been determined accordingly. Then our goal is to show that

$$\text{Goal: } \mathbb{P}_{\mathbf{X} \sim \mathcal{B}_\mathbf{p}}[ \ F(\Psi_{\mathbf{p},\mathbf{p}'}^\Lambda(\mathbf{X})) \leq \epsilon \ | \ F(H_{\boldsymbol{\beta}^*}(\mathbf{X})) \geq 1 - \delta \ ] \geq 1 - \epsilon$$

for sufficiently large $n$. Letting

$$X_i' = \frac{p_i - p_i'}{|p_i - p_i'|} X_i \quad (i = 1, \cdots, n)$$

,

$$\alpha = \mathbb{P}_{\mathbf{X} \sim \mathcal{B}_\mathbf{p}}[\beta_0 + \sum_{i=1}^n \beta_i X_i < 0]$$

and

$$\gamma = AC(\beta_0 + \sum_{i=1}^n \beta_i X_i)$$

, we get

$$\mathbb{P}_{\mathbf{X} \sim \mathcal{B}_\mathbf{p}}[ \ F(\Psi_{\mathbf{p},\mathbf{p}'}^\Lambda(\mathbf{X})) \leq \epsilon \ | \ F(H_{\boldsymbol{\beta}^*}(\mathbf{X})) \geq 1 - \delta \ ]$$

$$= \mathbb{P}_{\mathbf{X} \sim \mathcal{B}_\mathbf{p}}[ \ F(\Psi_{\mathbf{p},\mathbf{p}'}^\Lambda(\mathbf{X})) \leq \epsilon \ | \ F(\left|\beta_0 + \sum_{i=1}^n \beta_i X_i\right|) \leq \delta \ ]$$

$$\geq -\frac{\alpha}{\delta - \gamma} + \frac{1}{1 + (2\alpha + \gamma)/(\delta - \gamma)} \cdot \mathbb{P}_{\mathbf{X} \sim \mathcal{B}_\mathbf{p}}[ \ F(\Psi_{\mathbf{p},\mathbf{p}'}^\Lambda(\mathbf{X})) \leq \epsilon \ | \ F(\beta_0 + \sum_{i=1}^n \beta_i X_i) \leq \delta \ ]$$

$\quad$ (by $Lemma 3$)

$$= -\frac{\alpha}{\delta - \gamma} + \frac{1}{1 + (2\alpha + \gamma)/(\delta - \gamma)} \cdot \mathbb{P}_{\mathbf{X} \sim \mathcal{B}_\mathbf{p}}[ \ F(\Psi_{\mathbf{p},\mathbf{p}'}^\Lambda(\mathbf{X})) \leq \epsilon \ | \ F(\sum_{i=1}^n \beta_i X_i) \leq \delta \ ]$$

$$= -\frac{\alpha}{\delta - \gamma} + \frac{1}{1 + (2\alpha + \gamma)/(\delta - \gamma)} \cdot \mathbb{P}_{\mathbf{X} \sim \mathcal{B}_\mathbf{p}}[$$

$$F(\sum_{i=1}^n \frac{p_i - p_i'}{|p_i - p_i'|} \Lambda(p_i, p_i') X_i') \leq \epsilon \ | \ F(\sum_{i=1}^n \frac{p_i - p_i'}{|p_i - p_i'|} \beta(p_i, p_i') X_i') \leq \delta \ ] \quad \text{(by } equation \ 10\text{)}$$

$$\geq -\frac{\alpha}{\delta - \gamma} + \frac{1}{1 + (2\alpha + \gamma)/(\delta - \gamma)} \cdot (1 - \epsilon/2)$$

$\quad$ (by Lemma 6, provided that $n \geq N^{Lemma6}(\epsilon/2, M, \delta_{min})$)

$$\geq -\frac{\alpha}{\delta_{min} - \gamma} + \frac{1}{1 + (2\alpha + \gamma)/(\delta_{min} - \gamma)} \cdot (1 - \epsilon/2)$$

In the second last inequality, we could apply $Lemma 6$ provided that $n \geq N^{Lemma6}(\epsilon/2, M, \delta_{min})$ because

1. $\delta \in [\delta_{min}, \delta_0^{Lemma6}(\epsilon/2, M)]$

2. $\frac{p_i - p_i'}{|p_i - p_i'|} \Lambda(p_i, p_i') > 0$ (since $\Lambda$ is sign-respecting) and $\frac{1}{M} \leq |\Lambda(p_i, p_i')| \leq M$

3. $\frac{p_i - p_i'}{|p_i - p_i'|} \beta(p_i, p_i') > 0$ and $\frac{1}{M} \leq |\beta(p_i, p_i')| \leq M$

4. $\mathbf{X}' = (X_1', \cdots, X_n') \sim \mathcal{B}_{\mathbf{p}''}$, where $p_i'' = \begin{cases} p_i & (p_i > p_i') \\ 1 - p_i & (p_i < p_i') \end{cases} \in [\epsilon, 1-\epsilon]$ $(i = 1, \cdots, n)$.

Now, it is enough to show that the last quantity is $\geq 1 - \epsilon$ when $n \geq N'$ for some $N' = N'(\epsilon, \delta_{min})$. (Then, $N = \max(N', N^{Lemma6}(\epsilon/2, M, \delta_{min}))$ will satisfy the theorem statement) Since $\gamma \to 0$ as $n \to \infty$ in the convergence rate that depends only on $\epsilon$, it remains to show that the same is true for $\alpha$. Using equation 11 and equation 10, we get

$$
\begin{aligned}
\mathbb{E}(\beta_0 + \sum_{i=1}^{n} \beta_i X_i) = \beta_0 + \sum_{i=1}^{n}(2p_i - 1)\beta_i &= \sum_{i=1}^{n}(p_i - p_i')\beta_i \\
&= k(1 - 2\eta)\sum_{i=1}^{n}|p_i - p_i'|\,|\beta(p_i, p_i')| \\
&\geq k(1 - 2\eta)\frac{\epsilon}{M(\epsilon)}n
\end{aligned}
\tag{31}
$$

Therefore by Hoeffding's concentration inequality, since $\beta_i \in [-k(1-2\eta)M(\epsilon), k(1-2\eta)M(\epsilon)] = [a_i, b_i]$ $(i = 1, \cdots, n)$, we get

$$
\begin{aligned}
\alpha = \mathbb{P}[\beta_0 + \sum_{i=1}^{n}\beta_i X_i < 0] \\
\leq \mathbb{P}[\sum_{i=1}^{n}\beta_i X_i - \mathbb{E}[\sum_{i=1}^{n}\beta_i X_i] < -k(1-2\eta)\frac{\epsilon}{M(\epsilon)}n] \\
\leq \exp(-\frac{2(k(1-2\eta)\frac{\epsilon}{M(\epsilon)}n)^2}{\sum_{i=1}^{n}(b_i - a_i)^2}) \\
= \exp(-\frac{\epsilon^2}{2M(\epsilon)^4}n)
\end{aligned}
\tag{32}
$$

, finishing the proof.

### A.5.3 PROOF OF LEMMA 6

Now we will prove Lemma 6. We will make a bootstrapping argument composed of two stages: (1) proving the theorem imposing a certain restriction on $(p_1, \cdots, p_n, a_1, \cdots, a_n, b_1, \cdots, b_n)$ (2) proving the general case using the "restricted theorem".

First, let us prove the restricted theorem. The restriction will be clarified later on. Let $0 < \epsilon < 1/2$ and $M \geq 1$. Define

$$
\alpha = \Phi^{-1}(\epsilon/2)
$$

and

$$
\delta_0 = \Phi(M^2(\alpha - \Phi^{-1}(1 - \epsilon/2)))
$$

, where $\Phi$ is the cumulative density function of the standard normal distribution. We argue that $\delta_0$ satisfies the theorem statement. To show that, take arbitrary $\delta_{min} \in (0, \delta_0]$, $\delta \in [\delta_{min}, \delta_0]$, $\mathbf{p} = (p_1, \cdots, p_n) \in [\epsilon, 1-\epsilon]^n$ and $a_1, \cdots, a_n, b_1, \cdots, b_n \in [\frac{1}{M}, M]$. We have to show that

$$
\text{Goal: } \mathbb{P}_{\mathbf{X} \sim \mathcal{B}_{\mathbf{p}}}[F(\sum_{i=1}^{n} a_i X_i) \leq \epsilon \mid F(\sum_{i=1}^{n} b_i X_i) \leq \delta] \geq 1 - \epsilon
\tag{33}
$$

Define

$$
\mathbf{V}_i = \begin{pmatrix} a_i(X_i - \mu(p_i))/\sqrt{\sum_{i=1}^{n}a_i^2\sigma(p_i)^2} \\ b_i(X_i - \mu(p_i))/\sqrt{\sum_{i=1}^{n}b_i^2\sigma(p_i)^2} \end{pmatrix}, \quad (i = 1, \cdots, n)
\tag{34}
$$

and

$$
\mathbf{S} = \begin{pmatrix} S_1 \\ S_2 \end{pmatrix} = \sum_{i=1}^{n}\mathbf{v}_i
$$

, where $\mu(p_i) = \mathbb{E}[X_i]$ and $\sigma(p_i) = \mathbb{E}[(X_i - \mathbb{E}X_i)^2]$. Since $\mathbf{v}_i$ are independent and $\mathbb{E}[\mathbf{v}_i] = \begin{pmatrix} 0 \\ 0 \end{pmatrix}$, we can apply Lemma 4, provided that the covariance matrix $\Sigma = Cov[\mathbf{S}]$ is invertible. Indeed, we

may assume that $\Sigma$ is invertible because $\Sigma$ has the form

$$\Sigma = \begin{pmatrix} 1 & \kappa \\ \kappa & 1 \end{pmatrix}, \quad \kappa = \frac{\sum_{i=1}^{n} a_i b_i \sigma(p_i)^2}{\sqrt{\sum_{i=1}^{n} a_i^2 \sigma(p_i)^2} \sqrt{\sum_{i=1}^{n} b_i^2 \sigma(p_i)^2}} \tag{35}$$

and the conclusion of the theorem is trivial when $\kappa = 1$ (Since then $(a_1, \cdots, a_n) \sim (b_1, \cdots, b_n)$). The inverse is:

$$\Sigma^{-1} = \frac{1}{1 - \kappa^2} \begin{pmatrix} 1 & \kappa \\ \kappa & 1 \end{pmatrix} \tag{36}$$

Therefore, applying Lemma 4, we get, for $\mathbf{Z} \sim N(0, \Sigma)$,

$$|\mathbb{P}[\mathbf{S} \in U] - \mathbb{P}[\mathbf{Z} \in U]| \le C\gamma \text{ for any convex } U \subseteq \mathbb{R}^2 \tag{37}$$

, where $C$ is an absolute constant and

$$\gamma = \sum_{i=1}^{n} \mathbb{E}[\|\Sigma^{-1} \mathbf{V}_i\|_2^3]. \tag{38}$$

$\kappa$ has the following lower bound:

$$\kappa = \frac{\sum_{i=1}^{n} a_i b_i \sigma(p_i)^2}{\sqrt{\sum_{i=1}^{n} a_i b_i (\frac{a_i}{b_i}) \sigma(p_i)^2} \sqrt{\sum_{i=1}^{n} a_i b_i (\frac{b_i}{a_i}) \sigma(p_i)^2}} \ge \frac{1}{\sqrt{\max_{i=1}^{n} \frac{a_i}{b_i} \max_{i=1}^{n} \frac{b_i}{a_i}}} \ge M^{-2} \tag{39}$$

On the other hand, an upper bound comes from our "restriction".

$$\text{bootstrapping restriction: } \kappa \le 0.99 \tag{40}$$

Note that the case where $\kappa > 0.99$ should be, at least intuitvely, easier to prove, since the two events in equation 33 become more correlated in that case. We assume this upper bound temporarily to make sure $\gamma$ is bounded from above properly. From equation 36 and equation 40, it can be easily deduced that

$$\gamma \le \frac{C'}{\sqrt{n}}$$

for some $C' = C'(\epsilon, M)$.

Now, we're going to prove equation 33 in light of equation 37. To do so, note that the probability density function of $Z$ can be written as:

$$f_Z(z_1, z_2) = \frac{1}{2\pi \det \Sigma} \exp(-\frac{1}{2} \begin{pmatrix} z_1 \\ z_2 \end{pmatrix}^T \Sigma^{-1} \begin{pmatrix} z_1 \\ z_2 \end{pmatrix})$$

$$= \frac{1}{2\pi(1 - \kappa^2)} \exp(-\frac{1}{2(1 - \kappa^2)}(z_1^2 - 2\kappa z_1 z_2 + z_2^2))$$

Also, note that from this we can calculate:

$$\frac{\int_{-\infty}^{\alpha} f_Z(z_1, z_2) dz_1}{\int_{-\infty}^{\infty} f_Z(z_1, z_2) dz_1} = \Phi(\frac{\alpha - \kappa z_2}{\sqrt{1 - \kappa^2}}). \tag{41}$$

For sufficiently large $n$, we have

$$\mathbb{P}_{\mathbf{X} \sim \mathcal{B}_{\mathbf{p}}}[F(\sum_{i=1}^{n} a_i X_i) \leq \epsilon \mid F(\sum_{i=1}^{n} b_i X_i) \leq \delta]$$

$$= \mathbb{P}_{\mathbf{X} \sim \mathcal{B}_{\mathbf{p}}}[F(S_1) \leq \epsilon \mid F(S_2) \leq \delta]$$

$$= \mathbb{P}_{\mathbf{X}}[F(S_1) \leq \epsilon \text{ and } F(S_2) \leq \delta] \ / \ \delta \text{ (by Lemma 1)}$$

$$= \mathbb{P}_{\mathbf{X}}[\mathbb{P}_{\mathbf{X}'}[S_1' \leq S_1] \leq \epsilon \text{ and } \mathbb{P}_{\mathbf{X}'}[S_2' \leq S_2] \leq \delta] \ / \ \delta$$

$$\geq \mathbb{P}_{\mathbf{X}}[\mathbb{P}_{Z'}[Z_1' \leq S_1] \leq \epsilon - C\gamma \text{ and } \mathbb{P}_{Z'}[Z_2' \leq S_2] \leq \delta - C\gamma] \ / \ \delta \text{ (by equation 37)}$$

$$= \mathbb{P}_{\mathbf{X}}[S_1 \leq \Phi^{-1}(\epsilon - C\gamma) \text{ and } S_2 \leq \Phi^{-1}(\delta - C\gamma)] \ / \ \delta$$

$$\geq \mathbb{P}_{\mathbf{X}}[S_1 \leq \alpha \text{ and } S_2 \leq \Phi^{-1}(\delta - C\gamma)] \ / \ \delta$$

$$\geq -C\gamma/\delta + \mathbb{P}_Z[Z_1 \leq \alpha \text{ and } Z_2 \leq \Phi^{-1}(\delta - C\gamma)] \ / \ \delta \text{ (by equation 37)}$$

$$= -C\gamma/\delta + (1 - C\gamma/\delta) \cdot \frac{\int_{-\infty}^{\Phi^{-1}(\delta - C\gamma)}(\int_{-\infty}^{\alpha} f_Z(z_1, z_2) dz_1) dz_2}{\int_{-\infty}^{\Phi^{-1}(\delta - C\gamma)}(\int_{-\infty}^{\infty} f_Z(z_1, z_2) dz_1) dz_2}$$

$$\geq -C\gamma/\delta + (1 - C\gamma/\delta) \cdot \inf_{z_2 \leq \Phi^{-1}(\delta - C\gamma)} \frac{\int_{-\infty}^{\alpha} f_Z(z_1, z_2) dz_1}{\int_{-\infty}^{\infty} f_Z(z_1, z_2) dz_1}$$

$$= -C\gamma/\delta + (1 - C\gamma/\delta) \cdot \inf_{z_2 \leq \Phi^{-1}(\delta - C\gamma)} \Phi(\frac{\alpha - \kappa z_2}{\sqrt{1 - \kappa^2}}) \text{ (by equation 41)}$$

$$= -C\gamma/\delta + (1 - C\gamma/\delta) \cdot \Phi(\frac{\alpha - \kappa \Phi^{-1}(\delta - C\gamma)}{\sqrt{1 - \kappa^2}})$$

Now, let us further estimate the last term:

$$\Phi(\frac{\alpha - \kappa \Phi^{-1}(\delta - C\gamma)}{\sqrt{1 - \kappa^2}}) \geq \Phi(\frac{\alpha - \kappa \Phi^{-1}(\delta_0)}{\sqrt{1 - \kappa^2}})$$

$$= \Phi(\frac{\alpha - \kappa M^2(\alpha - \Phi^{-1}(1 - \epsilon/2))}{\sqrt{1 - \kappa^2}})$$

$$\geq \Phi(\frac{\alpha - (\alpha - \Phi^{-1}(1 - \epsilon/2))}{\sqrt{1 - \kappa^2}})$$

$$(\alpha - \Phi(1 - \epsilon/2) \text{ is negative} + equation \ 39)$$

$$\geq 1 - \epsilon/2$$

Therefore continuing the previous chain of inequalities,

$$\mathbb{P}_{\mathbf{X} \sim \mathcal{B}_{\mathbf{p}}}[F(\sum_{i=1}^{n} a_i X_i) \leq \epsilon \mid F(\sum_{i=1}^{n} b_i X_i) \leq \delta]$$

$$\geq -C\gamma/\delta + (1 - C\gamma/\delta) \cdot (1 - \epsilon/2)$$

$$\geq -C\delta^{-1}C'/\sqrt{n} + (1 - C\delta^{-1}C'/\sqrt{n}) \cdot (1 - \epsilon/2)$$

$$\geq -C\delta_{min}^{-1}C'/\sqrt{n} + (1 - C\delta_{min}^{-1}C'/\sqrt{n}) \cdot (1 - \epsilon/2)$$

$$\geq 1 - \epsilon$$

, when $n \geq N$ for some $N$ that depends only on $\epsilon$, $M$ and $\delta_{min}$. This finishes the proof with the restriction of equation 40.

Turning to the general case, let $0 < \epsilon < 1/2$ and $M \geq 1$. We take

$$\delta_0 = \min(\delta_0^{\text{bootstrap}}(\epsilon, M), \delta_0^{\text{bootstrap}}(\epsilon/2, 2M)).$$

We claim that $\delta_0$ satisfies the theorem statement. Let $\delta_{min} \in (0, \delta_0]$, $\delta \in [\delta_{min}, \delta_0]$, $\mathbf{p} = (p_1, \cdots, p_n) \in [\epsilon, 1 - \epsilon]^n$ and $\frac{1}{M} \leq a_1, \cdots, a_n \leq M$. Take

$$N = \max(N^{\text{bootstrap}}(\epsilon, M, \delta_{min}), N^{\text{bootstrap}}(\epsilon/2, 2M, \delta_{min})).$$

We may assume that

$$\kappa = \frac{\sum_{i=1}^{n} a_i b_i \sigma(p_i)^2}{\sqrt{\sum_{i=1}^{n} a_i^2 \sigma(p_i)^2} \sqrt{\sum_{i=1}^{n} b_i^2 \sigma(p_i)^2}} > 0.99$$

, since otherwise, we can rely on $\delta \in [\delta_{min}, \delta_0^{\text{bootstrap}}(\epsilon, M)]$ to conclude.

We will construct $\frac{1}{2M} \leq a_1', \cdots, a_n' \leq 2M$ and $\frac{1}{2M} \leq a_1'', \cdots, a_n'' \leq 2M$ such that

$$\kappa' = \frac{\sum_{i=1}^{n} a_i' b_i \sigma(p_i)^2}{\sqrt{\sum_{i=1}^{n} a_i'^2 \sigma(p_i)^2} \sqrt{\sum_{i=1}^{n} b_i^2 \sigma(p_i)^2}} \leq 0.99 \tag{42}$$

,

$$\kappa'' = \frac{\sum_{i=1}^{n} a_i'' b_i \sigma(p_i)^2}{\sqrt{\sum_{i=1}^{n} a_i''^2 \sigma(p_i)^2} \sqrt{\sum_{i=1}^{n} b_i^2 \sigma(p_i)^2}} \leq 0.99 \tag{43}$$

and

$$3a_i = a_i' + a_i'' \quad (1 \leq i \leq n) \tag{44}$$

. Then since $\delta \in [\delta_{min}, \delta_0^{\text{bootstrap}}(\epsilon/2, 2M)]$, we get

$$p' = \mathbb{P}[F(\sum_{i=1}^{n} a_i' X_i) \leq \epsilon/2 \mid F(\sum_{i=1}^{n} b_i X_i) \leq \delta] \geq 1 - \epsilon/2$$

and

$$p'' = \mathbb{P}[F(\sum_{i=1}^{n} a_i'' X_i) \leq \epsilon/2 \mid F(\sum_{i=1}^{n} b_i X_i) \leq \delta] \geq 1 - \epsilon/2$$

for $n \geq N$. Then we can finish the proof as follows:

$$\mathbb{P}[F(\sum_{i=1}^{n} a_i X_i) \leq \epsilon \mid F(\sum_{i=1}^{n} b_i X_i) \leq \delta]$$

$$= \mathbb{P}[F(\sum_{i=1}^{n} 3a_i X_i) \leq \epsilon \mid F(\sum_{i=1}^{n} b_i X_i) \leq \delta]$$

$$= \mathbb{P}[F(\sum_{i=1}^{n} (a_i' + a_i'') X_i) \leq \epsilon \mid F(\sum_{i=1}^{n} b_i X_i) \leq \delta]$$

$$\geq \mathbb{P}[F(\sum_{i=1}^{n} a_i' X_i) + F(\sum_{i=1}^{n} a_i'' X_i) \leq \epsilon \mid F(\sum_{i=1}^{n} b_i X_i) \leq \delta]$$

$$\geq \mathbb{P}[F(\sum_{i=1}^{n} a_i' X_i) \leq \epsilon/2 \text{ and } F(\sum_{i=1}^{n} a_i'' X_i) \leq \epsilon/2 \mid F(\sum_{i=1}^{n} b_i X_i) \leq \delta]$$

$$\geq p' + p'' - 1$$

$$\geq (1 - \epsilon/2) + (1 - \epsilon/2) - 1 = 1 - \epsilon$$

In the middle, we relied on the inequality $F(Y_1 + Y_2) \leq F(Y_1) + F(Y_2)$ where $Y_1 = \sum_{i=1}^{n} a_i' X_i$ and $Y_2 = \sum_{i=1}^{n} a_i'' X_i$, which can be easily checked.

Now it's enough to construct $a_i', a_i''$ that satisfy equation 42, equation 43 and equation 44. If we find $S \subseteq \{1, \cdots, n\}$ such that

$$\sum_{i \in S} a_i b_i \sigma(p_i) \simeq \frac{1}{2} \sum_{i=1}^{n} a_i b_i \sigma(p_i)^2 \tag{45}$$

and

$$\sum_{i \in S} a_i^2 \sigma(p_i) \simeq \frac{1}{2} \sum_{i=1}^{n} a_i^2 \sigma(p_i)^2 \tag{46}$$

, we can let

$$a_i' = \begin{cases} 2a_i & (i \in S) \\ a_i & (i \in S^c) \end{cases} \quad \text{and} \quad a_i'' = \begin{cases} a_i & (i \in S) \\ 2a_i & (i \in S^c) \end{cases}$$

, which leads to:

$$
\begin{aligned}
\kappa' &= \frac{\sum_{i=1}^{n} a_i' b_i \sigma(p_i)^2}{\sqrt{\sum_{i=1}^{n} a_i'^2 \sigma(p_i)^2} \sqrt{\sum_{i=1}^{n} b_i^2 \sigma(p_i)^2}} \\
&= \frac{2 \sum_{i \in S} a_i b_i \sigma(p_i)^2 + \sum_{i \in S^c} a_i b_i \sigma(p_i)^2}{\sqrt{4 \sum_{i \in S} a_i^2 \sigma(p_i)^2 + \sum_{i \in S^c} a_i'^2 \sigma(p_i)^2} \sqrt{\sum_{i=1}^{n} b_i^2 \sigma(p_i)^2}} \\
&\sim \frac{\sum_{i=1}^{n} (2 \cdot \frac{1}{2} + \frac{1}{2}) a_i b_i \sigma(p_i)^2}{\sqrt{(4 \cdot \frac{1}{2} + \frac{1}{2}) \sum_{i=1}^{n} a_i^2 \sigma(p_i)^2} \sqrt{\sum_{i=1}^{n} b_i^2 \sigma(p_i)^2}} \\
&= \frac{1.5}{\sqrt{2.5}} \kappa \le \frac{1.5}{\sqrt{2.5}} < 0.99
\end{aligned}
$$

$S$ that satisfies equation 45 and equation 46 can found, for example, by using a probabilistic argument. (Define a random set and show that it satisfies the desired properties with high probability using Hoeffding's concentration inequality)

## A.6   THE ERROR THEOREM

**Definition 9.** *A function $Err : \mathbb{R} \times \{-1, 1\} \to \mathbb{R}$ will be called a good error function if it satisfies the following properties:*

1. *For any $(\hat{y}_1, y_1), (\hat{y}_2, y_2) \in \mathbb{R} \times \{-1, 1\}$,*

$$
\hat{y}_1 y_1 > 0, \hat{y}_2 y_2 \le 0 \Rightarrow Err(\hat{y}_1, y_1) < Err(\hat{y}_2, y_2)
$$

2. *For any $\hat{y}_1, \hat{y}_2 \in \mathbb{R}$ and $y \in \{-1, 1\}$*

$$
\hat{y}_1 \ne \hat{y}_2 \Rightarrow Err(\hat{y}_1, y) \ne Err(\hat{y}_2, y)
$$

*That is,*

1. *Predictions that have the same sign with the target label always have lower error values than those that do not.*

2. *Different predictions on inputs that have the same target label result in distinguishable error values.*

**Theorem 2.** *Let $Err$ be a good error function. Given $0 < \epsilon < 1/2$ and $0 \le \eta < 1/2$, there exists $C = C(\epsilon, \eta) > 0$ such that for any $0 < \theta \le 1$, for sufficiently large $n$, the following holds:*

*For any combination of $\mathbf{p} = (p_1, \cdots, p_n)$ and $\mathbf{p}' = (p_1', \cdots, p_n')$ in $[\epsilon, 1 - \epsilon]^n$ with $|p_i - p_i'| \ge \epsilon$ $(1 \le i \le n)$, when $\boldsymbol{\beta}^* = (\beta_0, \beta_1 \cdots, \beta_n)$ is the risk-minimizing linear solution of $(\mathbf{X}, Y)$ for $(\mathbf{X}, Y, Z) \sim \mathcal{D}_{\mathbf{p}, \mathbf{p}'}^{\eta}$, we have*

$$
\mathbb{P}_{\mathbf{X}, Y, Z \sim \mathcal{D}_{\mathbf{p}, \mathbf{p}'}^{\eta}} [\; YZ = -1 \;|\; F(Err(\hat{Y}, Y)) \ge 1 - \theta \;] \ge \min(1, \frac{\eta}{\theta}) - \frac{C}{\sqrt{n}}
$$

*, where $\hat{Y} = \beta_0 + \sum_{i=1}^{n} \beta_i X_i$.*

## A.7   PROOF OF THEOREM 2

In proving Theorem 2, we tackle the different ranges of $\theta$ (1) $\theta > \eta$, (2) $\theta < \eta$ (3) $\theta = \eta$ separately. In all cases, we need the following facts:

- When

$$
\xi = \mathbb{P}[\hat{Y} Z \le 0]
$$

  , we have

$$
\xi \le \exp(-C_1 n) \tag{47}
$$

  for some $C_1 = C_1(\epsilon)$.

*Proof.* Use Hoeffding's inequality in conjunction with equation 11 and equation 10 in the same manner as in equation 31 and equation 32. $\qquad\square$

- When
$$\eta_0 = \mathbb{P}[\hat{Y}Y \le 0]$$
, we have
$$|\eta_0 - \eta| \le \xi \tag{48}$$

*Proof.*

$$
\begin{aligned}
|\eta_0 - \eta| &= \left| \mathbb{P}[\hat{Y}Y \le 0] - \mathbb{P}[YZ = -1] \right| \\
&\le \mathbb{P}[(\hat{Y}Y \le 0 \wedge YZ = 1) \vee (\hat{Y}Y > 0 \wedge YZ = -1)] \text{ (symmetric difference)} \\
&\le \mathbb{P}[\hat{Y}Z \le 0] \text{ (logical implication)} \\
&= \xi
\end{aligned}
$$

$\qquad\square$

- When
$$\gamma = AC(Err(\hat{Y}, Y)) \text{ (See Lemma 1)}$$
, we have
$$\gamma \le \frac{C_2}{\sqrt{n}} \tag{49}$$
for some $C_2 = C_2(\epsilon)$.

*Proof.*

$$
\begin{aligned}
\gamma &= \max_{x \in \mathbb{R}} \mathbb{P}[Err(\hat{Y}, Y) = x] \\
&\le \max_{x \in \mathbb{R}} (\mathbb{P}[Err(\hat{Y}, 0) = x] + \mathbb{P}[Err(\hat{Y}, 1) = x]) \\
&\le AC(Err(\hat{Y}, 0)) + AC(Err(\hat{Y}, 1)) \\
&\le AC(\hat{Y}) + AC(\hat{Y}) \text{ (Since } Err \text{ is "good")} \\
&= 2AC(\sum_{i=1}^{n} \beta_i X_i) \\
&\le \frac{C_2}{\sqrt{n}} \text{ (equation 10, Lemma 2)}
\end{aligned}
$$

$\qquad\square$

### A.7.1   THE CASE $\theta > \eta$

Assume $\theta > \eta$. From equation 48, equation 47 and equation 49, we see that
$$\theta \ge \eta_0 + \gamma \tag{50}$$
when $n$ is sufficiently large.

First, we argue that
$$\text{Claim: } \hat{Y}Y \le 0 \Rightarrow F(Err(\hat{Y}, Y)) \ge 1 - \theta. \tag{51}$$
Suppose otherwise. Then it happens with nonzero probability that $\hat{Y}Y \le 0$ and $F(Err(\hat{Y}, Y)) < 1 - \theta$. That is,
$$\exists (\hat{y}_1, y_1), \hat{y}_1 y_1 \le 0 \wedge F(Err(\hat{y}_1, y_1)) < 1 - \theta. \tag{52}$$
On the other hand, we have

$$
\begin{aligned}
\mathbb{P}[F(Err(\hat{Y}, Y)) &\ge 1 - \theta \wedge \hat{Y}Y > 0] \\
&\ge \mathbb{P}[F(Err(\hat{Y}, Y)) \ge 1 - \theta] - \eta_0 \\
&> \theta - \gamma - \eta_0 \text{ (by Lemma 1)} \\
&\ge 0 \text{ (by equation 50)}
\end{aligned}
$$

Since the event $F(Err(\hat{Y}, Y)) \geq 1 - \theta \wedge \hat{Y}Y > 0$ occurs with a nonzero probability, we have

$$\exists(\hat{y}_2, y_2), \hat{y}_2 y_2 > 0 \wedge F(Err(\hat{y}_2, y_2)) \geq 1 - \theta. \tag{53}$$

From equation 52 and equation 53, since $\hat{y}_1 y_1 \leq 0$, $\hat{y}_2 y_2 > 0$ but $Err(\hat{y}_1, y_1) < Err(\hat{y}_2, y_2)$, we find a contradiction with the definition of "good error function".

Now, we have

$$\mathbb{P}[YZ = -1 | F(Err(\hat{Y}, Y)) \geq 1 - \theta]$$
$$= \frac{\mathbb{P}[YZ = -1 \wedge F(Err(\hat{Y}, Y)) \geq 1 - \theta]]}{\mathbb{P}[F(Err(\hat{Y}, Y)) \geq 1 - \theta]]}$$
$$\geq \frac{\mathbb{P}[YZ = -1 \wedge \hat{Y}Y \leq 0]]}{\mathbb{P}[F(Err(\hat{Y}, Y)) \geq 1 - \theta]]} \text{ (by equation 51)}$$
$$\geq \frac{\mathbb{P}[YZ = -1 \wedge \hat{Y}Y \leq 0]]}{\theta} \text{ (by Lemma 1)}$$
$$= \frac{\mathbb{P}[YZ = -1 \wedge \hat{Y}Z \geq 0]]}{\theta}$$
$$\geq \frac{\mathbb{P}[YZ = -1] - \mathbb{P}[\hat{Y}Z < 0]]}{\theta} \geq \frac{\eta - \xi}{\theta}$$
$$\geq \frac{\eta}{\theta} - \frac{1}{\theta} \exp(-C_1 n) \text{ (by equation 47)}$$

### A.7.2 THE CASE $\theta < \eta$

Assume $\theta < \eta$. From equation 48 and equation 47, we see that

$$\theta < \eta_0 \tag{54}$$

when $n$ is sufficiently large.

First, we argue that

$$\text{Claim: } F(Err(\hat{Y}, Y)) \geq 1 - \theta \Rightarrow \hat{Y}Y \leq 0 \tag{55}$$

Suppose otherwise. Then it happens with nonzero probability that $F(Err(\hat{Y}, Y)) \geq 1 - \theta$ and $\hat{Y}Y > 0$. That is,

$$\exists(\hat{y}_1, y_1), \hat{y}_1 y_1 > 0 \wedge F(Err(\hat{y}_1, y_1)) \geq 1 - \theta. \tag{56}$$

On the other hand, we have

$$\mathbb{P}[F(Err(\hat{Y}, Y)) < 1 - \theta \wedge \hat{Y}Y \leq 0]$$
$$\geq \eta_0 - \mathbb{P}[F(Err(\hat{Y}, Y)) \geq 1 - \theta]$$
$$\geq \eta_0 - \theta \text{ (by Lemma 1)}$$
$$> 0 \text{ (by equation 54)}$$

Since the event $F(Err(\hat{Y}, Y)) < 1 - \theta \wedge \hat{Y}Y \leq 0$ happens with a nonzero probability, we have

$$\exists(\hat{y}_2, y_2), \hat{y}_2 y_2 \leq 0 \wedge F(Err(\hat{y}_2, y_2)) < 1 - \theta. \tag{57}$$

From equation 56 and equation 57, since $\hat{y}_1 y_1 > 0$, $\hat{y}_2 y_2 \leq 0$ but $Err(\hat{y}_2, y_2) < Err(\hat{y}_1, y_1)$, we find a contradiction with the definition of "good error function".

Now, we have

$$\mathbb{P}[YZ = -1 | F(Err(\hat{Y}, Y)) \geq 1 - \theta]$$

$$= \frac{\mathbb{P}[YZ = -1 \wedge F(Err(\hat{Y}, Y)) \geq 1 - \theta]]}{\mathbb{P}[F(Err(\hat{Y}, Y)) \geq 1 - \theta]]}$$

$$\geq \frac{\mathbb{P}[\hat{Y}Z > 0 \wedge F(Err(\hat{Y}, Y)) \geq 1 - \theta]]}{\mathbb{P}[F(Err(\hat{Y}, Y)) \geq 1 - \theta]]}$$

(equation 55, $\hat{Y}Y \leq 0 \wedge \hat{Y}Z > 0 \Rightarrow YZ \leq 0 \Rightarrow YZ = -1$)

$$\geq 1 - \frac{\xi}{\mathbb{P}[F(Err(\hat{Y}, Y)) \geq 1 - \theta]]}$$

$$\geq 1 - \frac{\xi}{\theta - \gamma} \text{ (by Lemma 1)}$$

$$\geq 1 - \frac{1}{\theta/2} \exp(-C_1 n) \text{ (by equation 47, equation 49)}$$

, when $n$ is sufficiently large.

### A.7.3 THE CASE $\theta = \eta$

The remaining case is when $\theta = \eta$. In this case, we rely on an approximation to the case $\theta < \eta_0$ (Note that the proof for the previous case $\theta < \eta$ relied on that $\theta < \eta_0$ when $n$ is sufficiently large.). Let

$$\eta_1 = \min(\eta, (1 - \exp(-n))\eta_0).$$

We get

$$\mathbb{P}[YZ = -1 | F(Err(\hat{Y}, Y)) \geq 1 - \eta]$$

$$\geq \mathbb{P}[YZ = -1 | F(Err(\hat{Y}, Y)) \geq 1 - \eta_1] \cdot \frac{\mathbb{P}[F(Err(\hat{Y}, Y)) \geq 1 - \eta_1]}{\mathbb{P}[F(Err(\hat{Y}, Y)) \geq 1 - \eta]}$$

$$\geq (1 - \frac{2}{\eta_1} \exp(-C_1 n)) \cdot \frac{\mathbb{P}[F(Err(\hat{Y}, Y)) \geq 1 - \eta_1]}{\mathbb{P}[F(Err(\hat{Y}, Y)) \geq 1 - \eta]}$$

(Applying the lower bound from the previous case)

$$\geq (1 - \frac{2}{\eta_1} \exp(-C_1 n)) \cdot \frac{\eta_1 - \gamma}{\eta} \text{ (Lemma 1)}$$

$$\geq (1 - \frac{4}{\eta} \exp(-C_1 n)) \cdot \frac{(1 - \exp(-n))\eta_0 - \gamma}{\eta}$$

$$\geq (1 - \frac{4}{\eta} \exp(-C_1 n)) \cdot \frac{(1 - \exp(-n))(\eta - \exp(-C_1 n)) - \frac{C_2}{\sqrt{n}}}{\eta}$$

(equation 48, equation 47, equation 49)

$$\geq 1 - \frac{2C_2/\eta}{\sqrt{n}}$$

, when $n$ is sufficiently large.

## B EXPERIMENTAL DETAILS

### B.1 SYNTHETIC DATASET

**Dataset details** The 2-D synthetic classification dataset consists of the majority groups (can be classified with the spurious features) and the minority groups (cannot be classified with the spurious features). The training set consists of 1,000 majority group samples and five minority group samples. Specifically, the majority group sample is sampled from the multivariate Gaussian distributions $\mathcal{N}([5.0, 5.0], 1.3\mathbf{I})$ and $\mathcal{N}([-5.0, -5.0], 1.3\mathbf{I})$ for positive and negative classes, respectively. On the contrary, the minority group sample is sampled from the $\mathcal{N}([5.0, -5.0], 1.3\mathbf{I})$ and

$\mathcal{N}([-5.0, 5.0], 1.3\mathbf{I})$. Notably, before we feed the training samples into the model, the second feature (invariant feature) is manually scaled down 1/10 times. This downscaling leads to relatively insignificant gradients for the invariant features. Thus, the invariant feature is hard to learn.

**Model details**   We use the simplest neural network architecture for this experiment: the fully connected neural network with a single hidden layer with ReLU activation. The number of the hidden neuron is 50,000. For the training, we use SGD optimization for all evaluated models. For the ERM, we use 1e-4 as the learning rate; 1e-5 as the weight decay. For the END's identification model, we use 5e-4 as the learning rates; 1e-5 as the weight decay; 1e-3 as the confidence regularization; 1e-2 as the p-value threshold. The SGLD weight sample is saved every 3,000 iterations. For the JTT and END's debiased model, we use 1e-5 as the learning rate; 5e-4 as the weight decay for both the identification model and final model. The hyperparameters for JTT and END's debiased model are determined based on the argument from Sagawa et al. (2019); Liu et al. (2021): the importance weighting approaches should use relatively lower learning rates and higher weight decays. Every evaluated method is trained in 100,000 iterations with batch-size 32, excepts for the identification model of END and JTT: 30,000 iterations. Additionally, we add the additional datasets (error set for JTT, SCF set for END) 30 times.

## B.2   GROUP ROBUSTNESS BENCHMARK DATASET

**Dataset details**   We use the two image classification datasets, the CelebA and Waterbirds. The two datasets have been used to evaluate the group robustness (worst-group accuracy). Each dataset has the group information (attributes, labels). And the attributes is the spurious-cue. For instance, the target class of the CelebA dataset is hair color, but the model can predict the class by abusing the spurious cue *"gender"*. This abuse leads to the degradation of accuracy for a certain group. If the model has group-robustness, this degradation is insignificant.

Here, we elucidate the input features, the group information, and the target classes of the dataset:

- **Waterbirds** (Wah et al., 2011): The input data is the image of the birds and their backgrounds. The target classes are either "waterbirds" or "landbirds". Notably, the dataset has the group information (background images, labels): "waterbirds with (water/land) background" or "landbirds (water/land) background". In the training dataset, most of the waterbird images have the water background, and vice-versa. Here, only 5% of the training dataset has a contradictory background (e.g., landbird on the water background). We consider 4 groups: [waterbird, water background], [landbird, water background], [waterbird, land background], and [landbird, land background].

- **CelebA** (Liu et al., 2015) The input data is the image of the celebrities. Similar to the Sagawa et al. (2019) and Liu et al. (2021), the target class is the hair color, "blond" or "not blond". Here, the spurious attributes are the "male" or "female." The label and spurious attributes are spuriously correlated like in the Waterbird case (e.g., most of the "male" group has the "not blond" label). We consider 4 groups: [blond, female], [not blond, female], [blond, male], and [not blond, male].

Moreover, we add symmetric noise to the two datasets. In practice, we randomly change the target labels of each sample with probabilities of [0, 10%, 20%, 30%].

**Model details**   In our experiments on the benchmark datasets, we follow the experimental setup of Liu et al. (2021) for LfF, ERM, and JTT. Specifically, we use the same model architecture for all evaluated methods: ResNet50 (He et al., 2016), pre-trained with the ImageNet. For the baseline approaches' hyperparameters, we follow the prior studies (Nam et al., 2020; Liu et al., 2021). Moreover, similar to the Liu et al. (2021), the model selection is based on the validation worst-group accuracy.

For the Waterbirds dataset, all evaluated models are trained with up to 300 epochs and batch-size 64, except for the identification model of END and JTT, 50 epochs. The SGD optimizers with the 0.9 momentum are used, except for the LfF. The model selection is based on the worst-group accuracy on the validation dataset for all methods. The ERM uses 1e-3 as the learning rate, 1e-4 as the L2 regularization; The JTT and END's debiased model uses 1e-5 as the learning rates, 1.0 as the L2

regularization, and adding the additional dataset for JTT (error set) 50 times and for END (SCF set) 100 times; The LfF uses the Adam optimizer, with 1e-4 learning rates and 1e-4 L2 regularization. The $q$ of the LfF is 1e-3; END uses 1e-3 as the learning rate, the 1e-4 as the L2 regularization, and 3e-1 as the confidence regularization for the identification model. We save the SGLD weight samples every 5 epochs.

For the CelebA dataset, the models are trained with up to 50 epochs and batch-size 64, except for the identification of END and JTT, 5 and 1 epochs respectively. Similar to the Waterbirds, the SGD optimizers with the 0.9 momentum are used, except for the LfF. The model selection is based on the worst-group accuracy on the validation dataset for all methods. The ERM uses 1e-4 as the learning rate, 1e-4 as the L2 regularization; The JTT uses 1e-5 as the learning rates, 1e-1 as the L2 regularization, and adding the additional dataset (error set of JTT, SCF set of END) 50 times; The LfF uses the Adam optimizer, with 1e-4 learning rates and 1e-4 L2 regularization. The $q$ of the LfF is 1e-3; END uses 5e-2 as the learning rate, the 1e-4 as the L2 regularization, and 1e-3 as the confidence regularization for the identification model. We save the SGLD weight samples every 1 epochs.

### B.3  DRUG-TARGET AFFINITY REGRESSION DATASET

**DTA task**   The Drug-Target Affinity (DTA) regression task is a significant task for early-stage drug discovery. We use two well-known benchmark datasets: Davis (Davis et al., 2011) and KIBA (Tang et al., 2014). The inputs of the datasets are the Simplified Molecular Input Line-entry System (SMILES) sequence—the sequence of the drug molecule—and the amino acid sequence—the sequence of the target protein. Thus, similar to other sequence-based deep learning tasks, the input data are the one-hot encoded sequences. The target value is real-valued drug-target affinity. The dataset has 5 different folds, which is suggested in (Öztürk et al., 2018). Thus, the results in Table 2 are average and standard deviation over 5 trials.

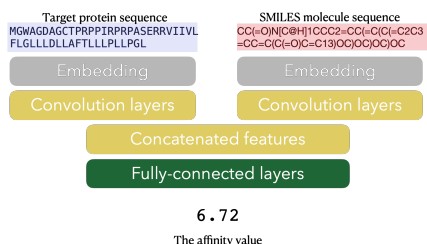

Figure 5: The schema of the DTA task with DeepDTA architecture.

- **Davis** dataset consists of clinically relevant kinase inhibitor ligands and their affinity values–dissociation constant $K_d$. As Öztürk et al. (2018) does, we rescale the target affinity value: $-\frac{\log K_d}{1e^6}$. The Davis dataset consists of 68 drugs and 442 target protein sequences, a total of 30,056 affinity values. The additional detail of the Davis dataset is that the target affinity value is 5 if the affinity is lower than 5 (Davis et al., 2011). This could be a noisy label that does not represent the true affinity value.

- **KIBA** dataset includes kinase protein sequences and SMILES molecule sequences, similar to the Davis. The KIBA dataset uses its own affinity score, the KIBA score. The dataset has 2,111 compounds and 229 proteins, a total of 118,254 affinity values.

**DeepDTA model**   We use the well-known simple, but effective DeepDTA architecture Öztürk et al. (2018). The DeepDTA consists of the one-dimensional convolution layers to encode the sequences and the fully-connected layers to output the affinity value from the concatenated latent features. Figure 5 illustrates the DeepDTA architecture.

**Details**   For the all evaluated model, we use 1e-3 as the learning rate; 1e-4 as the weight decay; 0.1 as the dropout probability; 256 as the batch size, except for the 1e-1 learning rate for the identification model. We train the models with 200 epochs. For the identification model, we train the model with 200 and 100 epochs for the Davis and KIBA, respectively. We save the SGLD samples every 20 and 10 epochs for the Davis and Kiba, respectively. The baseline "hard" pick up the top-500 high-loss samples for the oversampling.

Table 3: Worst-group accuracy (WG ACC) and average accuracy (ACC) of END with different loss functions and the regularization.

|  | Waterbirds 30% noise | | CelebA 30% noise | |
| --- | --- | --- | --- | --- |
|  | **AVG ACC** | **WG ACC** | **AVG ACC** | **WG ACC** |
| **JTT** | 0.012 (0.00) | 0.106 (0.01) | 0.151 (0.16) | 0.258 (0.12) |
| **END (w/o both)** | 0.787 (0.04) | 0.716 (0.04) | 0.698 (0.03) | 0.611 (0.06) |
| **END (w/o reg)** | 0.888 (0.02) | 0.570 (0.02) | 0.901 (0.01) | 0.757 (0.06) |
| **END (w/o MAE)** | 0.932 (0.01) | 0.459 (0.08) | 0.725 (0.04) | 0.618 (0.03) |
| **END** | 0.854 (0.01) | **0.818 (0.03)** | 0.892 (0.01) | **0.778 (0.03)** |

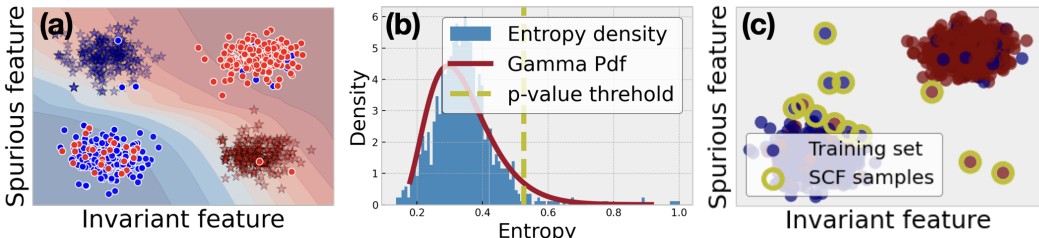

Figure 6: 2-D classification results of the *identification model* on synthetic data. **(a)** The prediction results. The colors represents the classes. The dots and translucent stars represents training and test data respectively; **(b)** the histogram of the predictive entropy and the corresponding Gamma distribution; **(c)** the detected SCF samples.

## C ADDITIONAL EXPERIMENTS

### C.1 ABLATION STUDY

In this subsection, we conduct the experiments on the CelebA and Waterbirds with 30% label noise to verify contributions of END's components (Table 3). Specifically, we train the models via the END framework with its variants: training the identification model without the MAE loss or confidence regularization, or both.

In this experiment, we have several implications. Firstly, utilizing uncertainty is a major factor in improving the worst-case group accuracy compared to exploiting the error (loss). In particular, compared to JTT, every END variant improves the worst group accuracy by at least 0.4. This supports one of our main claims: utilizing uncertainty is a proper approach to improving the group robustness under the noisy label scenarios. Secondly, the cooperation between the noisy-label robust loss and the overconfidence regularization significantly contributes to the group robustness in the presence of the label noise. In practice, the combination of both shows the outstanding worst group accuracy with noisy labels, as shown in Table 2 and 3. In contrast, without the overconfidence regularization (END w/o reg), only utilizing the noisy robust loss (MAE) does not show outstanding improvement compared to the proposed method (END). We interpret this degradation is due to the overconfident uncertainty for the minority group, as *Model-B* in Figure 2 shown. In addition, without the noisy label robust loss (END w/o reg and MAE), the worst group accuracy is degraded since the identification model memorizes the noisy labels, as we stated in the last two paragraphs of Sec 4. Hence, we conclude that the cooperation between the noisy robust MAE loss and the overconfidence regularization is the key component in the classification tasks, as we argued.

### C.2 QUALITATIVE EXPERIMENT

We conduct a qualitative experiment on the waterbird dataset to evaluate the characteristic of the SCF set. Specifically, we obtain two separate groups of images: **(1)** the most uncertain images that their uncertainty is estimated by identification model (Figure 7); (2) randomly chosen images (Figure 8).

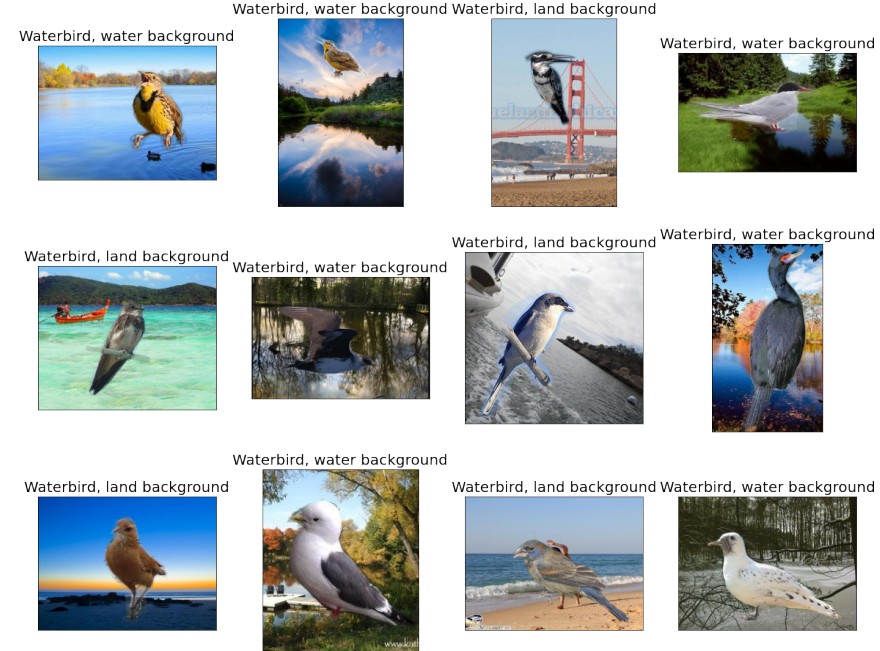

Figure 7: The images of the waterbirds dataset with high-predictive uncertainty (Top-12).

We observed several abnormal properties of the uncertain (SCF) images. For instance, the uncertain image has "land background" as the group label but has a water background or vice versa. In addition, several images have ambiguous backgrounds: hard to determine whether the water or land background is. In contrast, the randomly chosen images have relatively certain backgrounds. We believe that the ambiguous backgrounds of the SCF dataset allow that model focuses on the image of the bird—the true causality of the label.

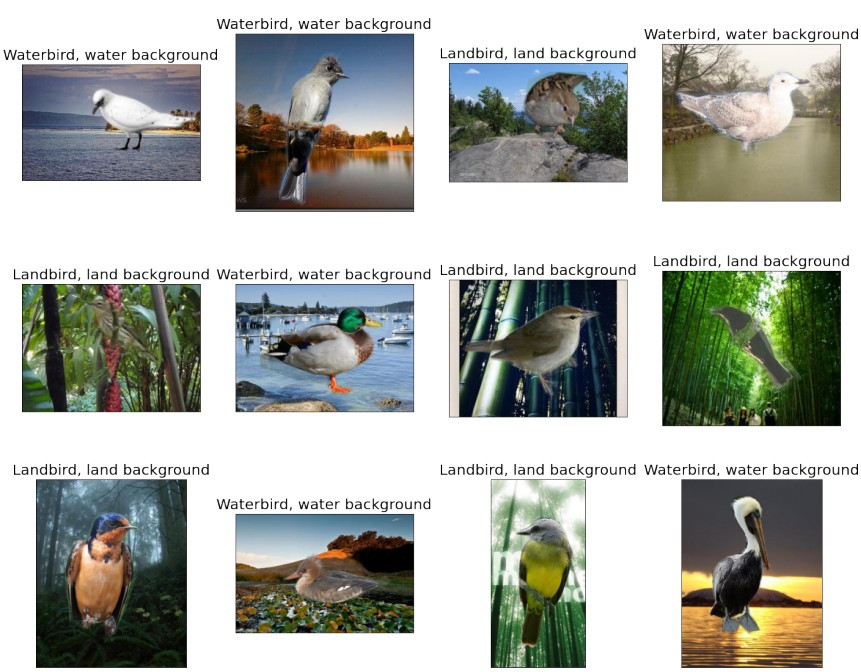

Figure 8: Randomly chosen images of the waterbird dataset.

