# OpenReview forum: "Improving group robustness under noisy labels using predictive uncertainty"
_ICLR.cc/2023/Conference — Submitted to ICLR 2023_

### Official Review · Reviewer_YCzi · 2022-10-20

**Confidence:** 3
**Correctness:** 3
**Technical Novelty And Significance:** 3
**Empirical Novelty And Significance:** 3
**Recommendation:** 5

**Clarity, Quality, Novelty And Reproducibility:**

The paper is well-motivated. However, some unclear descriptions hinder understanding of the paper fully.

**Strength And Weaknesses:**

Strengths:
1. The work focuses on addressing an important and challenging problem. The motivation behind the design choices is clear. All the related works are properly addressed.
2. Very simple method, which can be useful for practitioners.

Weaknesses and questions:
1. This work proves the predictive uncertainty is sufficient to identify SCF samples with label noise by assuming a very specific setting, and that the proof only considers the binary classification problem, while it remains unclear, even intuitively, how to extend it to true multi-class classification case.
2. One key to the problem is to distinguish between SCF samples and noise samples. Assume that the proof of the paper is correct and general that samples with high uncertainty are SCF samples, while samples with high uncertainty/large losses are also commonly considered to be noisy samples in noisy label learning. My question is what is the relationship between high entropy and large loss?
3. Some of the details about theoretical formalism are not clear to me. Equation 4 is difficult to understand and needs further explanation, and why beta in Theorem 1 is the solution of the regression?
4. For preventing overconfidency of networks, Section 5.1 proposes to use a regularizer to encourage a smaller predictive entropy, but the predictive entropy is also an indicator to discriminate SCF samples, so the reliable entropy is also desirable. How to trade off these two points?


**Summary Of The Paper:**

This paper aims to improve the worst-case performance in the present of label noise. Because typical high-loss-based approaches for atypical groups of the data may fail under this setting, the paper introduces an uncertainty-based approach to identify spurious-cue-free (SCF) samples, and then trains a model with the SCF set oversampled dataset. This paper tries to theoretically prove the uncertainty is a proper indicator of the SCF samples, and experiments show effectiveness.

**Summary Of The Review:**

I am not sure whether the proposed method is well supported. If the concerns are properly addressed I am willing to increase my score.

---

> ### Author Response · Authors · 2022-11-19
> **Responses to the questions and concerns related to the theoretical results**
>
> We appreciate the reviewer’s careful attention to our theoretical results. The following answers respond to the questions and concerns:
>
> - **This work proves the predictive uncertainty is sufficient to identify SCF samples with label noise by assuming a very specific setting, and that the proof only considers the binary classification problem, while it remains unclear, even intuitively, how to extend it to true multi-class classification case**
>
>
> As you mentioned, our theoretical analysis does assume a specific setting -  **binary** **classification** with a **linear model**, under a **specific data generation process.** However, we want to note that (1) “binary classification with a linear model” is a *common* assumption in theoretical analyses of prior works (2) the specific data generation process was for the most part essential to convey the main points of our claims. We elaborate on each of these points:
>
> - Prior works on the spurious-cue problem, such as [Nagarajan et al., 2020] and [Sagawa et al., 2020], undertook theoretical analyses under the setting of **binary classification with a linear model**.
> - The most peculiar aspect of our data generation process was that each class (out of two) has a probability of having a certain value for each feature ($x_i$). However, this setting was essential in defining our SCS (Spurious-Cue Scoring) function, the criteria for spurious samples. This is because the aforementioned probabilities were used to quantify to what degree a sample’s feature conforms to the dataset-level correlations.
>
> Nevertheless, we do admit that it could have been better if our theoretical analysis covered the multi-class case as well, although prior works have the same limitation. Also, as you mentioned, it is not clear how the current formulation and proof extend to that case. This can be explored in our future work.
>
> - **One key to the problem is to distinguish between SCF samples and noise samples. Assume that the proof of the paper is correct and general that samples with high uncertainty are SCF samples, while samples with high uncertainty/large losses are also commonly considered to be noisy samples in noisy label learning. My question is what is the relationship between high entropy and large loss?**
>
>
> Our answer to your last question is directly linked to the main claim of our paper: while “large loss” cannot distinguish “SCF (Spurious-Cue Free) samples” from “samples with noisy label”, “high entropy” can. This claim can be deconstructed into the following sub-claims:
>
> (Claim 1) For a random sample, having a “large loss” implies having a “noisy label” with a high probability (certainly higher than the dataset-level noise ratio).
>
> (Claim 2) For a random sample, having a “high entropy” implies that it is an “SCF sample” with a very high probability.
>
> (Claim 3) For a random sample, having a “high entropy” does not increase the probability of having a “noisy label”.
>
> In the paper, we backed these claims (assuming an ideal situation) in the following places:
>
> (1) Claim 1 is stated and proved in Theorem 2. (Appendix A.6)
>
> (2) Claim 2 is stated and proved in Theorem 1. (Section 4 and Appendix A.4)
>
> (3) Claim 3 is mentioned in Section 4, where we stated “the probability of the samples with a high predictive uncertainty to have a label noise is not greater than $\eta$, the original label noise ratio”. The reasoning is in the previous sentence: “the selectivity of the predictive entropy is independent of the presence of the label noise (whether $yz = −1$)”. Essentially, this is true due to the (plausible) assumption that the appearance of label noise is independent to the “input” $x$, based on which the predictive entropy is computed.
>
> - **Some of the details about theoretical formalism are not clear to me. Equation 4 is difficult to understand and needs further explanation, and why beta in Theorem 1 is the solution of the regression?**
>
> The upper line of Equation 4 reads, “With high probability (at least $1-\epsilon$), samples $(z,x)$ with top-$\delta$ predictive entropy (with respect to the solution $\beta^*$) has bottom-$\epsilon$ SCS (Spurious Cue Scoring) function value within the distribution associated to the ground-truth $z$.” Note that $\beta^*$ being the solution is not a conclusion, but a definition. One hint for an interpretation like this, is that the cumulative distribution function in the conditional event denotes the rank of $H_{\beta^*}(x)$ (in a real number in $[0, 1]$) within the data distribution. Similarly, $R(x,z)$ in the lower line is composed of two cumulative distribution functions of the form $F(\Psi(x))$, so it denotes the rank of the sample’s SCS function ($\Psi(x)$), within an appropriate distribution. The “appropriate distribution” depends on which one of the indicator functions is activated.

---

> > ### Comment · Reviewer_YCzi · 2022-11-28
> > **Thanks for the response**
> >
> > Thanks the authors for the comprehensive reply.
> >
> > I know that binary classification is often used as a simplification of multiclass classification when the analysis of multiclass case is tricky. However, I would still expect to see at least empirical observations to support the main motivation, for example, the dynamics of loss and entropy of SCF samples and noisy samples (with or without regularisation) during the training.
> > Why wouldn't the regularizer prevent the model from overfitting the noise? If so, the noisy samples would also have low entropy. If not, we know that neural networks learn patterns before they remember noise, does this mean that the model will learn SCF samples after it remembers noise?

---

> > > ### Author Response · Authors · 2022-12-12
> > > **Response to the Reviewer YCzi**
> > >
> > > ### Empirical observation of multi-class classification
> > >
> > > We agree with your point that the multi-class case might need empirical justifications. We conducted an experiment using synthetic data and obtained positive results. We will include results like these in Appendix A in our camera-ready submission.
> > >
> > > Essentially, we attempted to show that the conclusions of Theorem 1 and 2 may extend to multi-class cases. To do so, we sampled a noisy dataset from a data distribution similar to the one prescribed in the **Problem setup** paragraph of Section 4, fitted a model, and examined the relationships between the quantities *loss*, *predictive entropy* and *Spurious-Cue Scoring* (*SCS) function value* for the samples in the dataset.
> > >
> > > Most of the settings were the same as those of the Theorems, except that:
> > >
> > > - 5-class classification was considered instead of binary classification
> > > - A finite number of samples (50000) were fitted, instead of the distribution itself
> > > - Logistic regression was used instead of linear regression
> > > - A specific version of SCS function that suits the multi-class scenario was used. More specifically, we used the following multi-class equivalent of the function $s$ in **Definition 1: $s_i(p_1,\cdots,p_5)= \frac{p_i}{\sum_{j=1}^{5}p_j}-\frac{1}{5}$**
> > >
> > > Here are additional settings:
> > >
> > > - Number of features: 50
> > > - Base noise rate: 0.1
> > >
> > > Through 100 trials with different random seeds (randomness being in the dataset generation process), we could observe that
> > >
> > > 1. **The predictive entropy and the SCS function values had strong negative correlations (−0.75±0.07).**
> > > 2. **Noisy samples were evenly spread across the spectrum of predictive entropy.**
> > > 3. **Among the noise-free samples, the loss values and the SCS function values had negative correlations (−0.67±0.06)**
> > > 4. **Samples with high loss values were predominantly from noisy samples. From a visualization using a scatter plot, the distinction between the noisy and noise-free samples in terms of the loss values was clearly visible. In fact, the best threshold for this distinction resulted in only 26.7±18.3 exceptions, out of the 50000 samples (around 5000 of which were noisy).**
> > >
> > > Note that the first observation is in the line with the statement of Theorem 1, that samples with the lower predictive entropy are likely spurious-cue-free, and the fourth observation is aligned with the statement of Theorem 2.

---

> > > ### Author Response · Authors · 2022-12-12
> > > **Response to the Reviewer YCzi (2)**
> > >
> > > ### Concern about the regularization
> > >
> > > We interpret that your concern is one of: (1) it can be a problem that our regularization does not prevent the model from overfitting the noise, or (2) the regularizer would prevent the model from overfitting the noise, which can be problematic because the entropy of noisy samples can be increased.
> > >
> > > Unfortunately, we are not sure which interpretation is your intention. In the below comments, we reply assuming each case separately.
> > >
> > > **Interpretation #1 (Our regularization does not prevent the overfitting the noise)**
> > >
> > > In fact, our confidence regularization already prevents overfitting the noise. The original study of the confidence regularization [1] claimed and demonstrated that the regularization could prevent over*fitting* (by reducing over*confidence*). Therefore, we expect that—as the reviewer mentioned—the regularization can also prevent the model overfits the noisy label. Thus, the decision boundary is not much complex as in the case of Figure 2 (both for models A and B). This leads to the confidence of the noisy label remaining high (thus, the identification does not pick the noisy label).
> > >
> > > However, in the real-world case (e.g., much more high dimensional space than in the Figure 2 cases) the entropy of the noisy label can also be high, as we discussed in the last two paragraphs in Section 4. In the below comment (Interpretation #2), we argue that our framework may remain effective, even if the entropies of the noisy samples are high.
> > >
> > > **Interpretation #2 (The regularizer would prevent the model from overfitting the noise, which can be problematic)**
> > >
> > > As you mentioned, the regularizer could increase the entropy of noisy samples. However, we do not think this would deteriorate the effectiveness of our proposed method, at least through rendering the noisy samples over-sampled via the entropy criteria.
> > >
> > > In particular, we believe it is likely that the following holds:
> > >
> > > $(*)\quad \mathbb{P}[\text{being a noisy sample } | \text{ having a high entropy}]\simeq \text{(the base noise rate)}$
> > >
> > > , especially when the regularizer prevents the model from overfitting any samples, including the noisy samples.
> > >
> > > The main point of (*) is that the END framework with the confidence regularization does not make the identification model select noise samples excessively (the above conditional probability is not higher than the background noise level). Note that this point was also emphasized in Figure 4 of our paper. We can derive (*) when we assume the following two conditions:
> > >
> > > - Condition 1. Removing one data point from the training dataset may alter the trained parameters (hence the entropy function) only negligibly. This condition is realistic when the regularizer prevents the model from overfitting. Note that, as we mentioned in Concern #1, the confidence regularization makes Condition 1 more realistic since it prevents overfitting.
> > > - Condition 2. The presence of label noise in each sample is independent of (1) all other samples, and (2)  the "input" part of the sample at hand.
> > >
> > > The derivation of (*) goes:
> > >
> > > $$
> > > \mathbb{P}[(x,y)\text{ is noisy | } H_{\theta(\mathcal{D})}(x)\text{ is high}] \simeq \mathbb{P}[(x,y)\text{ is noisy | } H_{\theta(\mathcal{D}\setminus (x,y))}(x)\text{ is high}] = \mathbb{P}[(x,y)\text{ is noisy}] = \text{(the base noise rate)}
> > > $$
> > >
> > > where θ(D)\theta(\mathcal{D}) refers to the parameters obtained as a result of training on the dataset D\mathcal{D}. Here,
> > >
> > > - the first equality ($\simeq$) is from Condition 1, and
> > > - the second equality is from Condition 2.
> > >
> > > We hope these comments help you to clarify our method and relieve your major concern.
> > >
> > > (References)
> > >
> > > [1] Pereyra, G., Tucker, G., Chorowski, J., Kaiser, L., & Hinton, G. (2017). Regularizing Neural Networks by Penalizing Confident Output Distributions.

---

> ### Author Response · Authors · 2022-11-19
> **The purpose of using the regularizer**
>
> The following answer responds to the remaining concern of the reviewer:
>
> - **For preventing overconfidency of networks, Section 5.1 proposes to use a regularizer to encourage a smaller predictive entropy, but the predictive entropy is also an indicator to discriminate SCF samples, so the reliable entropy is also desirable. How to trade off these two points?**
>
> In fact, the purpose of using the regularizer lies precisely in **enhancing the reliability** of the predictive entropy in the discrimination of SCF samples. Therefore, the way we see it, there is no trade-off. Without this, according to [1][2], neural networks tend to make **over-confident** predictions, which may give SCF samples mistakenly high confidence (low entropy). Therefore, the regularizer aims to mitigate this problem by **encouraging large predictive entropy** (beware that Equation 6 is the sign-reversed entropy). Even with this regularizer, highly obvious samples (such as those with spurious-cues) would have low predictive entropy.
>
> [1] Hein, M., Andriushchenko, M., & Bitterwolf, J. (2019). Why relu networks yield high-confidence predictions far away from the training data and how to mitigate the problem. In *Proceedings of the IEEE/CVF Conference on Computer Vision and Pattern Recognition*
>  (pp. 41-50).
>
> [2] Guo, C., Pleiss, G., Sun, Y., & Weinberger, K. Q. (2017, July). On calibration of modern neural networks. In *International conference on machine learning*
>  (pp. 1321-1330). PMLR.

---

### Official Review · Reviewer_ptnQ · 2022-10-24

**Confidence:** 4
**Correctness:** 3
**Technical Novelty And Significance:** 3
**Empirical Novelty And Significance:** 2
**Recommendation:** 3

**Clarity, Quality, Novelty And Reproducibility:**

* For clarity, the paper is well-written and easy-to-follow.
* For novelty, the authors propose an interesting and novel idea, but the technical quality is not very satisfactory in terms of experiments and diving deep about uncertainty.
* As mentioned in weakness, the paper does not provide the information about reproducibility.


**Strength And Weaknesses:**

Strengths:
* Practical problem settings for noisy label environments.
* Theoretical analysis on the proposed idea of leveraging predictive uncertainty to improve group robustness.
* Experiments on both synthetic and real-world benchmark datasets with improvements over baselines.

Weaknesses:
* The proposed method does not consider the characteristics of uncertainty like aleatoric and epistemic uncertainty. This could be important to figure out whether the uncertainty on specific predictions identifies the proper SCF samples.
* Although the proposed framework outperforms most of the baseline methods in worst-group accuracy, it generally performs worse than baseline methods in average accuracy, especially when the noises are fewer. The authors should elaborate on this phenomenon and justify the trade-off sacrifice.
* The paper lacks qualitative analysis. For example, the authors should demonstrate some examples of identified SCF samples.
* The authors do not provide the material for reproducibility or mention further plans.


**Summary Of The Paper:**

In this paper, the authors propose to leverage the predictive uncertainty to build an identification model, thereby discovering SCF samples and further improving the group robustness under the noisy label environment. They first demonstrate some theoretical justifications about how SCF samples can be obtained through computing predictive uncertainty. To eliminate the drawback of simply using ERM, the authors further propose the END framework. To identify the SCF samples, they train a Bayesian neural network based on MAE and confidence regularization. The debiased model can then be trained by considering both SCF samples and the original dataset. Experiments are conducted on a 2-d synthetic dataset, two group robustness benchmark datasets, and two DTA regression datasets. The experimental results demonstrate that the proposed framework END can obtain better WG accuracy and MSE.

**Summary Of The Review:**

In sum, I would recommend “3: reject, not good enough” because the paper is not ready for both of the framework properly using uncertainty and satisfactory experimental results.

---

> ### Author Response · Authors · 2022-11-19
> **Disentanglement of the different types of uncertainty would be good to explore in the future. All other concerns are addressed. References are listed in the next comment.**
>
> We sincerely appreciate the reviewer’s constructive feedback on the uncertainty characteristics and other suggestions. We respond to the major concerns and suggestions below:
>
> - **The proposed method does not consider the characteristics of uncertainty like aleatoric and epistemic uncertainty. This could be important to figure out whether the uncertainty on specific predictions identifies the proper SCF samples**
>
> **(Considering the types of uncertainties**) We truly understand your concern since the aleatoric uncertainty represents the “noise” of the data. Ideally, the perfectly disentangled aleatoric and epistemic uncertainty can improve our method. However, in the real world, it could be challenging, although some of the related research have shown positive results on this (e.g., [1]): the aleatoric and epistemic uncertainty can interact and thus leading to unreliable aleatoric uncertainty [2].
>
> Despite this limitation, we expect that the disentanglement between the two uncertainties could be great future work. In fact, we mentioned that only utilizing the epistemic uncertainty for END would be the proper future work in the Discussion section. Currently, we consider that the naive solution to improve this is considering the model proposed by [3], which models heteroscedastic aleatoric uncertainty by changing output layers as the gaussian parameters. Or, we could consider the evidential neural networks [4][5], which can naturally disentangle the aleatoric and the epistemic uncertainties, although their estimated uncertainty itself could be non-robust [6].
>
> - **Although the proposed framework outperforms most of the baseline methods in worst-group accuracy, it generally performs worse than baseline methods in average accuracy, especially when the noises are fewer. The authors should elaborate on this phenomenon and justify the trade-off sacrifice.**
>
> **(Average accuracy issue)** As we mentioned in the response to the question of the reviewer r2Mi, the main goal of the most group-robustness papers is improving the worst-group accuracy.  Our work also aims to improve the worst-group accuracy (but under the noisy labels), similar to other studies focusing on the worst-group accuracy. In this line of research, the degradation of the average accuracy is commonly condoned. For instance, the concurrent study [8] achieves the outstanding worst-group accuracy despite the degraded average accuracy. As our introduction section stated, the motivation behind mainly considering the worst-group accuracy is that the model could fail for a certain group of samples (under distribution shift) in spite of the high average accuracy. In fact, in terms of the average accuracy, the ERM—the *vanilla* method—generally shows the best performance, although the failure of ERM under distribution shift is common [7].
>
>
>
> - **The paper lacks qualitative analysis. For example, the authors should demonstrate some examples of identified SCF samples.**
>
> (**Qualitative experiment**) Similar to the qualitative analysis of the related study, the JTT, we add the additional qualitative analysis on the waterbirds dataset in Appendix C-2. In particular, we compare the randomly chosen images and the most uncertain images (by the identification model). The results show that the uncertain images have some abnormal images. For instance, the image is marked as the “land background,” but the majority of its background is water, unlike the randomly chosen images. We believe this type of data allows the model to only focus on the images of birds, not on their background.
>
> - **The authors do not provide the material for reproducibility or mention further plans.**
>
> (**Further plan**) Our further plans, which only utilize epistemic uncertainty and exploit more advanced methods to obtain a better uncertainty, are mentioned in the Discussion section.
>
> (**Reproducibility**) Since the policy of ICLR2023 does not allow putting the GitHub repository, we decided not to attach the repository link. Alternatively, we submit our codes on the openreview as supplementary materials.

---

> ### Author Response · Authors · 2022-11-19
> **References for the previous comment**
>
> [1] Mukhoti, J., Kirsch, A., van Amersfoort, J., Torr, P. H., & Gal, Y. (2021). Deep Deterministic Uncertainty: A Simple Baseline. *arXiv e-prints*, arXiv-2102.
>
> [2] Valdenegro-Toro, M., & Mori, D. S. (2022, June). A Deeper Look into Aleatoric and Epistemic Uncertainty Disentanglement. In *2022 IEEE/CVF Conference on Computer Vision and Pattern Recognition Workshops (CVPRW)* (pp. 1508-1516). IEEE.
>
> [3] Kendall, A., & Gal, Y. (2017). What uncertainties do we need in bayesian deep learning for computer vision?. *Advances in neural information processing systems*, *30*.
>
> [4] Sensoy, M., Kaplan, L., & Kandemir, M. (2018). Evidential deep learning to quantify classification uncertainty. *Advances in neural information processing systems*, *31*.
>
> [5] Charpentier, B., Borchert, O., Zügner, D., Geisler, S., & Günnemann, S. (2021, September). Natural Posterior Network: Deep Bayesian Predictive Uncertainty for Exponential Family Distributions. In *International Conference on Learning Representations*.
>
> [6] Kopetzki, A. K., Charpentier, B., Zügner, D., Giri, S., & Günnemann, S. (2021, July). Evaluating robustness of predictive uncertainty estimation: Are Dirichlet-based models reliable?. In *International Conference on Machine Learning* (pp. 5707-5718). PMLR.
>
> [7] Koh, P. W., Sagawa, S., Marklund, H., Xie, S. M., Zhang, M., Balsubramani, A., ... & Liang, P. (2021, July). Wilds: A benchmark of in-the-wild distribution shifts. In *International Conference on Machine Learning* (pp. 5637-5664). PMLR.
>
> [8] Nam, J., Kim, J., Lee, J., & Shin, J. (2021, September). Spread Spurious Attribute: Improving Worst-group Accuracy with Spurious Attribute Estimation. In *International Conference on Learning Representations.*

---

### Official Review · Reviewer_71MP · 2022-10-24

**Confidence:** 2
**Clarity, Quality, Novelty And Reproducibility:** This paper is well-written and easy t…
**Correctness:** 4
**Technical Novelty And Significance:** 4
**Empirical Novelty And Significance:** 4
**Recommendation:** 5

**Strength And Weaknesses:**

Strengths:
This paper is well-written and easy to follow. The main idea and modtivation is well-presented.
Weaknesses:
1. I don't think some of the author's claims are accurate enough. For example, the authors argue that an advantage of the proposed model is that it can be extended to regression models. However, other methods such as GroupDRO or JTT can also be extended to regression models with simple modifications.
2. The baseline used in the paper is much lower than that in the JTT paper. For example, on the Waterbirds dataset, the worst-case acc reported by the JTT paper is 86.7% and in this paper it is 84.6%.
3. Some very relevant other papers have not been cited and discussed[1][2]. [1] distinguishes noisy samples in long-tail data, which is relevant to the task of this paper.
[2] also uses uncertainty to solve the subpopulation shift problem. The way it builds uncertainty is similar to the snapshots mentioned in this paper.
[1] Identifying Hard Noise in Long-Tailed Sample Distribution
[2] UMIX: Improving Importance Weighting for Subpopulation Shift via Uncertainty-Aware Mixup
Minor:
At the end of page ten, citations to the paper "Enhancing the reliability of out-of-distribution image detection in neural networks" appear twice.


**Summary Of The Paper:**

This paper proposes a predictive uncertainty based method to identify spurious-cue-free samples from a training set with noisy labels. Then a new training dataset is by oversampling the SCF samples to train the debiased model. Separating minority group samples and noisy samples is indeed a trickier problem. But this paper seems to solve this problem well.

**Summary Of The Review:**

Overall I think this is a good paper, but it can still be improved.

---

> ### Author Response · Authors · 2022-11-19
> **The "inaccurate" claim is elaborated on. The reported performance metrics are justified. We will cite the proposed studies during camera-ready submission.**
>
> We appreciate the reviewer’s effort, especially suggested interesting concurrent and relevant works. We respond to the major concerns and questions of the reviewer below:
>
> - **I don't think some of the author's claims are accurate enough. For example, the authors argue that an advantage of the proposed model is that it can be extended to regression models. However, other methods such as GroupDRO or JTT can also be extended to regression models with simple modifications-**
>
> **(Extension to regression)** JTT cannot be directly used for the regression problems. For the JTT, the “Error set” consists of the wrongly classified samples, having a different predicted class compared to the ground truth label. However, for the regression problems, the *error set* cannot be defined, although we can manually choose the top-K highest loss samples for the error set as a proxy. In this context, due to such limitation, we design the baseline “Hard”. The “Hard” baseline builds the error set by picking the top-K highest loss samples and oversampling them. This “Hard” baseline is compared with the END framework in Sec 6.3. As a result, the END outperforms the “Hard.”
>
> - **The baseline used in the paper is much lower than that in the JTT paper. For example, on the Waterbirds dataset, the worst-case acc reported by the JTT paper is 86.7% and in this paper it is 84.6%**
>
> **(Different values of baseline performance)** We consider that the difference between the original paper and ours is a simple coincidence. Specifically, as Table 1 shows, the standard deviation of the worst-group accuracy for 3 different trials is almost 0.02 (2% std). This standard deviation is enough to take into account of the difference in the worst-group accuracy values between our work and the original paper. Furthermore, we observed that the worst-group accuracies of JTT on the Waterbirds datasets differ, according to the concurrent studies: 86.0% (UMIX [1]), and 83.8% (CNC [2]).
>
> - **Some very relevant other papers have not been cited and discussed[1][2]. [1] distinguishes noisy samples in long-tail data, which is relevant to the task of this paper. [2] also uses uncertainty to solve the subpopulation shift problem. The way it builds uncertainty is similar to the snapshots mentioned in this paper. [1] Identifying Hard Noise in Long-Tailed Sample Distribution [2] UMIX: Improving Importance Weighting for Subpopulation Shift via Uncertainty-Aware Mixup Minor: At the end of page ten, citations to the paper "Enhancing the reliability of out-of-distribution image detection in neural networks" appear twice**
>
> **(Relevant works)** We appreciate your suggestion of relevant works. However, [1] consider the “label imbalance,” unlike our work which tries to resolve the spurious cue problem. For instance, experiments in [1] do not include any group-robustness algorithms like JTT as the baselines. However, in terms of the methodology itself—simultaneously considering the label noise and ‘label’ shift—, [1] is related to our work. We will add this study to the related work during camera-ready submission.
>
> In addition, [2] the UMIX is the concurrent work that will be announced in NeurIPS2022 (after Nov 28). This work also looks fascinating since the work combines widely used Mixup with the importance weighting and utilizes the uncertainty for such importance weighting. We will conduct and analyze the experiments for the UMIX on the noisy label data.
>
> Thanks to the suggestion, we will discuss and cite these studies—exploiting uncertainties to resolve the distribution shifts or the label imbalance—in the related works section.
>
> [1] Han, Z., Liang, Z., Yang, F., Liu, L., Li, L., Bian, Y., ... & Yao, J. (2022). UMIX: Improving Importance Weighting for Subpopulation Shift via Uncertainty-Aware Mixup. *arXiv preprint arXiv:2209.08928*.
>
> [2] Zhang, M., Sohoni, N. S., Zhang, H. R., Finn, C., & Ré, C. (2022). Correct-N-Contrast: A Contrastive Approach for Improving Robustness to Spurious Correlations. *arXiv preprint arXiv:2203.01517*.

---

### Official Review · Reviewer_r2Mi · 2022-10-25

**Confidence:** 2
**Correctness:** 3
**Technical Novelty And Significance:** 3
**Empirical Novelty And Significance:** 2
**Recommendation:** 5

**Clarity, Quality, Novelty And Reproducibility:**

In general, the paper is clear and easy to understand. To the best of my knowledge, this is a novel work to some extent. The quality of this paper can be further improved through better empirical analysis, which at the moment is not quite comprehensive and significant.

**Strength And Weaknesses:**

Strength:

1. The paper is well-written and easy to follow in general.
2. The authors provide theoretical justification for their findings. Although I did not check through the theorem-proof, the findings are interesting.
3. The findings in Section 4 are interesting.

Weaknesses and Questions:

1. The comparison in Table 1 can be made more comprehensive. I suggest comparing with a few recent works such as SSA [1], and CnC [2]. Further, for CelebA, I see a large decline in the average accuracy, although the improvement in worst-group is marginal compared to JTT.
2. Is there any table reporting the accuracies of the identification model? I also wonder if the training time of END is significantly higher than the other compared methods.
3. It would be interesting to see the performance of END against spuriously correlated NLP datasets such as CivilComments, MultiNLI.

[1] Nam, J., Kim, J., Lee, J. and Shin, J., 2022. Spread Spurious Attribute: Improving Worst-group Accuracy with Spurious Attribute Estimation. ICLR 2022.

[2] Zhang, M., Sohoni, N.S., Zhang, H.R., Finn, C. and Ré, C., 2022. Correct-N-Contrast: A Contrastive Approach for Improving Robustness to Spurious Correlations. ICML 2022.

**Summary Of The Paper:**

This paper tackles the problem of spurious correlations. Specifically, the authors argue that the assumption that high-loss samples correspond to a lack of spurious cues is not always true. To this end, the authors show theoretically that using predictive uncertainty to identify spurious-cue-free samples is a better indicator. Based on this finding, they propose Entropy based Debiasing (END) to mitigate the problem of learning spurious correlations.

**Summary Of The Review:**

With the findings above, I currently give the paper a borderline reject score.

---

> ### Author Response · Authors · 2022-11-19
> **We performed the suggested experiments where we could. Our main claim remains unaffected by the concerns raised about performance.**
>
> We sincerely appreciate your thorough review, especially your suggestion for stronger baselines. We respond to your questions below.
>
> - **The comparison in Table 1 can be made more comprehensive. I suggest comparing with a few recent works such as SSA [1], and CnC [2]. Further, for CelebA, I see a large decline in the average accuracy, although the improvement in worst-group is marginal compared to JTT.**
>
> **(Issue on average accuracy)** The primary goal of group-robustness models, such as ours, is to improve the worst-case accuracy rather than the average accuracy. On the contrary, ERM only considers the average accuracy, resulting in inaccuracies to a specific group as stated in our introduction section and other studies [1][2]. Due to these different focuses, even the most recent study (SSA), introduced by Reviewer r2Mi, shows inferior average accuracy when compared to ERM (and JTT).
>
> **(Additional baseline to compare)** Thanks to the suggestion on the additional baselines. We conducted additional experiments with SSA, the stronger one among them. The experimental result shows that END still outperforms the baselines under the noisy label, which is our main goal.
>
> |  | Clean |  | 10% noise |  | 20% noise |  | 30% noise |  |
> | --- | --- | --- | --- | --- | --- | --- | --- | --- |
> | Waterbirds | WG Acc | Acc | WG Acc | Acc | WG Acc | Acc | WG Acc | Acc |
> | ERM | 0.687 (0.01) | 0.969 (0.00) | 0.648 (0.03) | 0.945 (0.01) | 0.649 (0.05) | 0.913 (0.01) | 0.629 (0.06) | 0.893 (0.03) |
> | ERM (GCE) | 0.674 (0.01) | 0.968 (0.00) | 0.665 (0.03) | 0.945 (0.00) | 0.651 (0.04) | 0.902 (0.00) | 0.660 (0.07) | 0.885 (0.03) |
> | LfF | 0.710 (0.03) | 0.947 (0.02) | 0.710 (0.00) | 0.914 (0.03) | 0.726 (0.03) | 0.858 (0.04) | 0.660 (0.04) | 0.899 (0.02) |
> | JTT | 0.851 (0.02) | 0.891 (0.01) | 0.359 (0.08) | 0.425 (0.09) | 0.025 (0.02) | 0.113 (0.01) | 0.012 (0.00) | 0.106 (0.01) |
> | SSA | 0.887 (0.01) | 0.918 (0.00) | 0.872 (0.02) | 0.885 (0.02) | 0.803 (0.02) | 0.825 (0.02) | 0.747 (0.02) | 0.773 (0.02) |
> | END | 0.828 (0.01) | 0.934 (0.01) | 0.831 (0.02) | 0.916 (0.01) | 0.828 (0.02) | 0.886 (0.03) | 0.826 (0.02) | 0.855 (0.03) |
>
> |  | Clean |  | 10% noise |  | 20% noise |  | 30% noise |  |
> | --- | --- | --- | --- | --- | --- | --- | --- | --- |
> | CelebA | WG Acc | Acc | WG Acc | Acc | WG Acc | Acc | WG Acc | Acc |
> | ERM | 0.487 (0.03) | 0.952 (0.00) | 0.477 (0.01) | 0.927 (0.01) | 0.480 (0.02) | 0.891 (0.01) | 0.485 (0.01) | 0.858 (0.03) |
> | ERM(GCE) | 0.502 (0.03) | 0.956(0.00) | 0.524 (0.02) | 0.950 (0.00) | 0.522 (0.02) | 0.941 (0.00) | 0.526 (0.04) | 0.920 (0.01) |
> | LfF | 0.788 (0.03) | 0.871 (0.04) | 0.080 (0.06) | 0.217 (0.01) | 0.027 (0.02) | 0.089 (0.02) | 0.052 (0.06) | 0.236 (0.25) |
> | JTT | 0.822 (0.02) | 0.915 (0.01) | 0.748 (0.02) | 0.810 (0.01) | 0.245 (0.36) | 0.357 (0.29) | 0.151 (0.16) | 0.258 (0.12) |
> | SSA | 0.899 (0.00) | 0.906 (0.01) | 0.735 (0.01) | 0.811 (0.00) | 0.674 (0.01) | 0.767 (0.01) | 0.632 (0.02) | 0.729 (0.01) |
> | END | 0.817 (0.01) | 0.887 (0.03) | 0.797 (0.01) | 0.881 (0.04) | 0.800 (0.04) | 0.896 (0.00) | 0.758 (0.09) | 0.901 (0.02) |
>
> The results show that, despite the outstanding worst-group accuracy of SSA on the clean dataset, SSA underperforms END on the noisy datasets, except for the waterbirds with 10% noise. We believe that the outperformance of END over SSA is remarkable since **(1)** the SSA has shown SOTA worst-group accuracy on the various benchmark dataset, including Waterbirds and CelebA datasets; **(2)** the SSA utilizes the additional group-information—a small amount of the group-labeled data—during training.
>
> - **Is there any table reporting the accuracies of the identification model? I also wonder if the training time of END is significantly higher than the other compared methods.**
>
> **(Accuracy of the identification model)** We checked that the performance of the identification model is poor. (See the below table for the performance) However, it does not harm the performance of the END framework itself since the identification model is used only to obtain the SCF samples.
>
> | Waterbirds | Acc | WG acc | CelebA | Acc | WG acc |
> | --- | --- | --- | --- | --- | --- |
> | clean | 0.944 | 0.134 | clean | 0.867 | 0 |
> | 10% noise | 0.944 | 0.129 | 10% noise | 0.867 | 0 |
> | 20% noise | 0.943 | 0.123 | 20% noise | 0.867 | 0 |
> | 30% noise | 0.94 | 0.111 | 30% noise | 0.867 | 0 |
>
> **(Training time of END)** The training time of the END is not significant compared with the other baselines such as JTT. Although we utilize the Bayesian neural network—specifically, SGLD—for the identification model, the trained epochs are almost identical to the thereof JTT.
>
> - **It would be interesting to see the performance of END against spuriously correlated NLP datasets such as CivilComments, MultiNLI.**
>
> Currently, we could not finish the experiment on the NLP datasets for the time limit. However, we agree that those additional experiments would make our work more interesting.

---

### Decision · Program_Chairs · 2023-01-20

**Decision:**

Reject

**Justification For Why Not Higher Score:**

As above, the reviewers noted multiple critiques of the work, primarily around apparently poor performance on noise-free settings compared to prior work, missing baselines for the empirical comparison, missing citations to relevant prior work, unclear details in the theory, and limitation of the proof to the binary classification setting. The author response clarified some points, but reviewers still maintained concerns on the first and last point in particular.

**Justification For Why Not Lower Score:**

N/A

**Metareview: Summary, Strengths And Weaknesses:**

The paper proposes a technique for coping with spurious correlations in the presence of label noise. A key observation is that prior works that rely on identifying high-loss samples may fail in such settings.

All reviewers raised critiques of the work, primarily around apparently poor performance on noise-free settings compared to prior work, missing baselines for the empirical comparison, missing citations to relevant prior work, unclear details in the theory, and limitation of the proof to the binary classification setting. The author response clarified some points, but reviewers still maintained concerns on the first and last point in particular.

From my reading, I concur with the reviewer sentiment that the work could have some interesting ideas, but the current execution falls a bit short. We thus do not think the paper suitable for publication at present, but certainly encourage the authors to submit their work to some future venue.

**Summary Of Ac-Reviewer Meeting:**

N/A